# Hamsters with long COVID present distinct transcriptomic profiles associated with neurodegenerative processes in brainstem

Anthony Coleon [1], Florence Larrous [1], Lauriane Kergoat [1], Magali Tichit[2], David Hardy [2], Thomas Obadia[3,4], Etienne Kornobis [3,5], Hervé Bourhy [1] & Guilherme Dias de Melo [1] ✉

Following infection with SARS-CoV-2, patients may experience with one or more symptoms that appear or persist over time. Neurological symptoms associated with long COVID include anxiety, depression, and memory impairment. However, the exact underlying mechanisms are not yet fully understood. Using golden hamsters as a model, we provide further evidence that SARS-CoV-2 is neuroinvasive and can persistently infect the brain, as viral RNA and replicative virus are detected in the brainstem 80 days after the initial infection. Infected hamsters exhibit a neurodegenerative signature in the brainstem, characterized by overexpression of innate immunity genes, and altered expression of genes involved in the dopaminergic and glutamatergic synapses, in energy metabolism, and in proteostasis. These infected animals exhibit persistent depression-like behavior, impaired short-term memory, and late-onset signs of anxiety. Finally, we provide evidence that viral and immunometabolic mechanisms coexist in the brainstem of SARS-CoV-2-infected hamsters, contributing to the manifestation of neuropsychiatric and cognitive symptoms.

COVID-19 remains a contemporary global public health issue, with more than 776 million cases reported to date, including more than 7 million deaths[1]. After the acute phase, not all COVID-19 survivors have complete resolution of symptoms, even several months after the first clinical signs. It's estimated that 6.5-12% of patients may exhibit persistent symptoms, such as fatigue, anosmia, respiratory problems, cognitive and psychosocial distress, as part of a new entity commonly called long COVID or post-acute sequelae of COVID-19 (PASC)[2–4]. Long Covid symptoms may be experienced regardless of the severity of initial symptoms, even though some risk factors to develop long Covid have been identified, which include sex, age, body weight, and preexisting respiratory affections[5–7].

SARS-CoV-2 has multiple targets; the infection affects not only the respiratory tract, but also the central nervous system (CNS), and may affect neurons, glial cells, and the immune metabolism of the brain[8–11]. The infection has been associated with neurological disease[12], can exacerbate brain aging[13], and trigger neurodegeneration[14], which can lead to long-term cognitive deficits and mood disorders[4,15]. In a recent study involving 231 French patients suffering from long Covid, 91.8% of them were still experiencing neurological and neurocognitive symptoms one year after their first consultation at a long Covid clinic[4].

Among the possible causes of long COVID, viral persistence is a key issue. It has been described that replicating SARS-CoV-2 can persist in bronchioalveolar lavage macrophages for more than six months after infection in non-human primates[16]. There is also evidence that SARS-CoV-2 can persist in various body compartments, including the brain[17] even after the virus has been cleared from the respiratory tract[18].

[1]Institut Pasteur, Université Paris Cité, Lyssavirus Epidemiology and Neuropathology Unit, Paris, France. [2]Institut Pasteur, Université Paris Cité, Histopathology Core Facility, Paris, France. [3]Institut Pasteur, Université Paris Cité, Bioinformatics and Biostatistics Hub, Paris, France. [4]Institut Pasteur, Université Paris Cité, G5 Infectious Diseases Epidemiology and Analytics, Paris, France. [5]Institut Pasteur, Université Paris Cité, Plate-forme Technologique Biomics, Paris, France. ✉e-mail: guilherme.dias-de-melo@pasteur.fr

Viral persistence in the brain and a pro-inflammatory status may lead to altered energetic and neurotransmitter metabolisms[19–21] all possible factors that could induce the manifestation of neuropsychiatric and cognitive symptoms[10,22–24]. However, the exact neuroanatomic basis and molecular mechanisms of these signs is not yet defined.

Here we show that SARS-CoV-2 infects and persists in the brainstem for at least 80 days after the intranasal inoculation in hamsters. The infected animals exhibit a neurodegenerative molecular signature in the brainstem, which is associated with an active infection and an inflammatory response. The late phase is primarily characterized by dysregulated genes associated to neurotransmission and altered energy metabolism. All of these changes in the brain, which share some characteristics with neurodegenerative diseases, may be the underlying causes of prolonged and persistent neurobehavioral changes, including signs of anxiety, depression, and memory loss. Altogether, we describe the hamster model for long COVID and provide conclusive evidence that neuropsychiatric symptoms and cognitive impairment may follow the acute infection. This work uncovers the long-standing effects of SARS-CoV-2 infection on brain metabolism and behavior.

## Results

### Hamsters infected with SARS-CoV-2 exhibit different clinical profiles in acute and in late phases, with sex- and variant-dependent effects

We performed an 80-days follow-up study (Fig. 1A) to characterize long-term effects of the infection in golden hamsters intranasally-inoculated with the ancestral SARS-CoV-2 Wuhan and the Delta and Omicron/BA.1 variants. We first investigated the differences in the clinical picture induced by the different viruses in a time- and sex-dependent manner (Fig. 1B–E). Wuhan and Delta induced a significant body weight loss (Dunn's test $p = 0.0003$ and $p = 0.0173$, respectively) and high clinical scores in male hamsters as 4 days post-infection (dpi) (Fig. 1B, C). Female hamsters infected with Wuhan and Delta also showed significant weight loss (Dunn's test $p = 0.0099$ and $p = 0.0048$, respectively), but milder clinical scores compared to males (Mann-Whitney test $p = 0.0054$) (Fig. 1D, E). Animals infected with Omicron/BA.1, however, presented no or very limited weight loss and lower clinical scores compared to animals infected with the other SARS-CoV-2 variants (Fig. 1B–E). The peak of the acute phase of the infection was defined as 4 dpi, time-point where the animals presented the highest clinical score (Fig. 1D, E) and severe airways pathology, exemplified by destruction of the olfactory epithelium and lung inflammation (Supplementary Fig. 1).

No more clinical sign was noticeable after 10 dpi. From 10 dpi until 80 dpi, the infected animals exhibited a progressive recovery from weight loss; however, their body weights remained lower than those of the mock-infected animals (Fig. 1B, C). Notably, among the males, the infected animals did not reach the weight of the control group by the end of the experiment (Fig. 1B). At the time of sampling (80 dpi), all animals were considered clinically healthy, even if the lung weight-to-body weight ratio (LW/BW) was slightly high in some animals (Supplementary Fig. 1H, I).

### SARS-CoV-2 infects and persist in the brainstem of intranasally-inoculated hamsters

We then investigated the presence and persistence of the different SARS-CoV-2 variants in the brainstem of intranasally-inoculated hamsters. To this aim, we initially performed a follow-up study to quantify the viral load in the brainstem at different time-points after infection of male and female hamsters with SARS-CoV-2 Wuhan. Remarkably, genomic and sub-genomic viral RNA were detected in the brainstem as early as four hours after the intranasal inoculation, with stable and elevated levels at 1-, 2- and 4-dpi, regardless of the sex. Genomic RNA was still detected at 14-, 30- and 80-dpi, but in

contrast, sub-genomic RNA was below the limit of detection (Supplementary Fig. 2A, C).

Next, we measured the viral RNA loads in the brainstem of male and female animals infected with Wuhan and the variants Delta and Omicron/BA.1 in two time-points: at 4 dpi and 80 dpi. During the acute phase at 4 dpi, SARS-CoV-2 genomic RNA was detected in all samples, regardless of the sex, the variant, or the clinical presentation (Fig. 1F). Sub-genomic RNA was found in all Wuhan-infected samples, and in an less consistent manner for the Delta- and Omicron/BA.1-infected samples (Fig. 1F). The viral titers in the brainstem were positive, albeit at a notably low level ($<10^2$ TCID50/mL). Nevertheless, we were able to isolate and amplify infectious virus from the brainstems of 100% of Wuhan-infected animals (4/4 males and 4/4 females), 87.5% of Delta (3/4 males and 4/4 females), and 100% of Omicron/BA.1 (4/4 males and 4/4 females) at 4 dpi (Fig. 1G). The presence of infectious virus was determined by the detection of the viral spike and nucleoprotein, as well as by the induction of syncytia in the infected cell cultures, which occurred two to four days after infection of the cells. (Fig. 1H, Supplementary Fig. 3A).

Remarkably, we still detected viral genomic RNA in the brainstem of the animals at 80 dpi (Fig. 1I). Although sub-genomic RNA was below the limit of detection, indicating low viral load, we were able to isolate and amplify infectious SARS-CoV-2 from the brainstem of infected hamsters at 80 dpi (Fig. 1K, Supplementary Fig. 3B). In these samples, the viral titers were remarkably low ($<10^2$ TCID50/mL), yet we observed cytopathic effect and detected viral proteins five to seven days after infection of the cells. We amplified infectious virus from the brainstems of 75% of Wuhan-infected animals (3/4 males and 3/4 females), 87.5% of Delta (4/4 males and 3/4 females), and 75% of Omicron/BA.1 (3/4 males and 3/4 females) at 80 dpi (Fig. 1J). In parallel, we also quantified the viral RNA loads in the airways (nasal turbinates and lungs) and in other parts of the brain than the brainstem (olfactory bulbs, cerebral cortex, cerebellum) of the animals at 80 dpi. Genomic SARS-CoV-2 RNA was detected in the airways of all animals. In the other brain areas, however, the detection was variable (Supplementary Fig. 4), and no sub-genomic viral RNA was detected.

### SARS-CoV-2 induces distinct phenotypic glial patterns in the brainstem of hamsters in acute and late infection

Having demonstrated the persistence of SARS-CoV-2 RNA and infectious virus in the brainstem of infected hamsters, we next analyzed the histopathological changes at 4 and 80 dpi, focusing on the olfactory bulbs (considered the entry point of the virus into the brain) and on the brainstem (Fig. 2A–C). In general, the histopathological analysis of the brain revealed no relevant microscopic alterations neither during the acute nor during the late phase of the infection (Supplementary Fig. 5). We also quantified the levels of serum neurofilament light chain (NfL) to assess neuroaxonal damage in male and female hamsters intranasally-inoculated with Wuhan at 80 dpi, but observed no difference between infected and mock-infected animals (Supplementary Fig. 2B, D).

To better assess the status of glial cells in the olfactory bulbs and in the brainstem, we used GFAP (glial fibrillary acidic protein) to identify astrocytes (Fig. 2D–R) and IBA-1 (ionized calcium-binding adapter molecule 1) to identify microglial cells (Supplementary Fig. 5P-AD) by immunohistochemistry. The astrocytic involvement during SARS-CoV-2 presented an interesting pattern, with high detection of GFAP (astrogliosis) frequently noticed in the olfactory bulbs, and in subpial and perivascular regions in the thalamus and in the pons of infected males at 4 dpi (Fig. 2I–M). A notable finding was the occurrence of subpial, perivascular, and subependymal astrogliosis at 80 dpi (Fig. 2O–R). We observed a significant increase in GFAP staining in the midbrain and medulla oblongata of infected animals (Fig. 2S,T).

Regarding microglia, we observed foci of reactive microglial cells in the olfactory bulbs (Supplementary Fig. 5U) in both

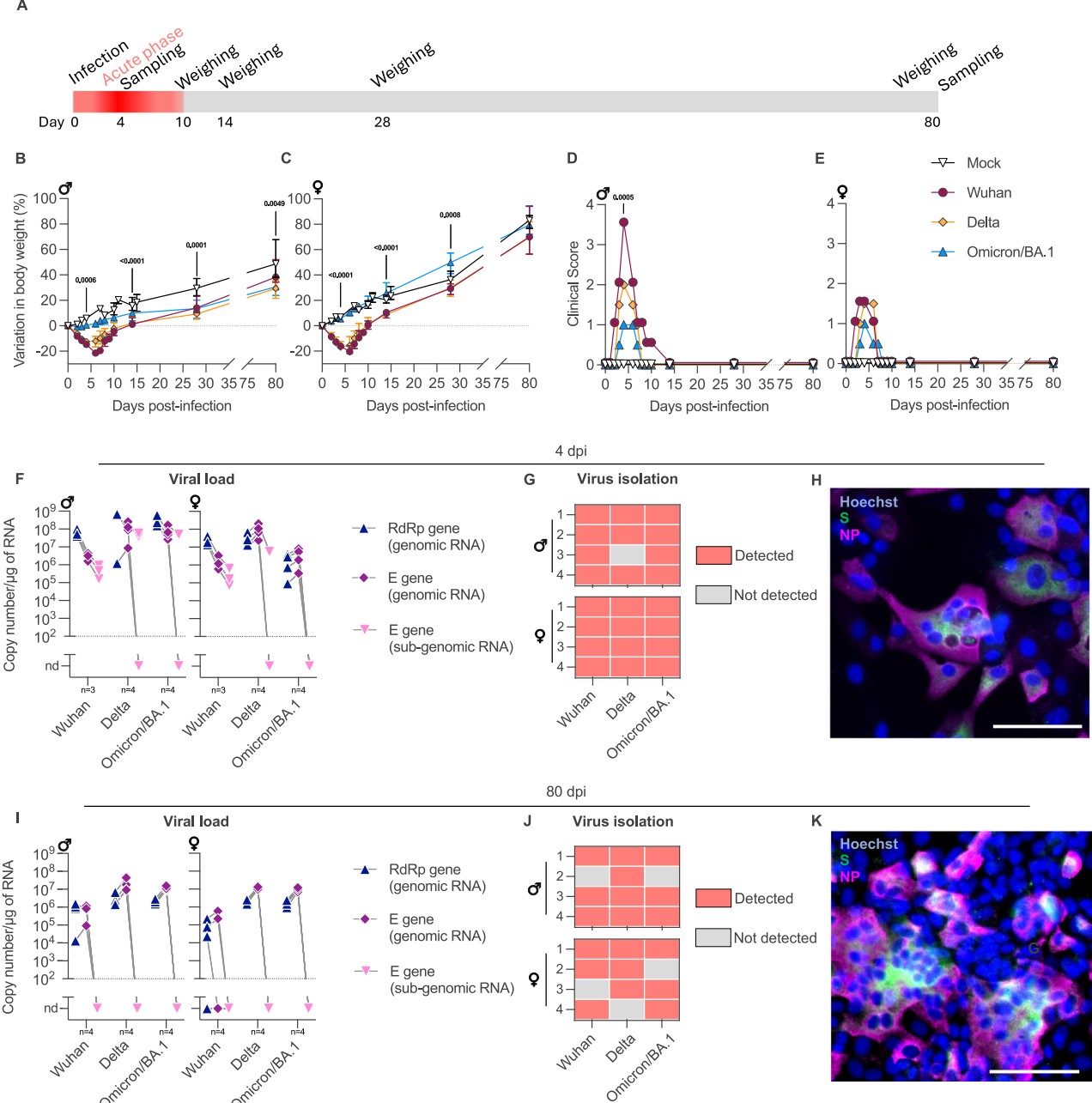

**Fig. 1 | Long-term clinical profile and brainstem viral load in hamsters intranasally-inoculated with SARS-CoV-2 Wuhan or the variants Delta and Omicron/BA.1. A** General experimental outline. **B**, **C** Body weight variation in male and female hamsters over 80 days post-infection (dpi). Horizontal lines indicate the median and the interquartile range. Kruskal-Wallis test performed at 4, 14, 28, and 80 dpi (the $p$ value is shown if $p < 0.05$). **D**, **E** Clinical score over 80 dpi. The clinical score is based on a cumulative 0-4 scale: ruffled fur, slow movements, apathy, and absence of exploration activity. Horizontal lines indicate the median and the interquartile range. Kruskal-Wallis test performed at 4 dpi (the $p$ value is shown if $p < 0.05$). **B**–**E** Between 0 and 28 dpi (mock-infected: $n = 20$ males + 18 females; Wuhan: $n = 12$ males + 12 females; Delta: $n = 4$ males + 4 females; Omicron/BA.1: $n = 4$ males + 4 females). At 80 dpi (mock-infected: $n = 12$ males + 12 females; Wuhan: $n = 8$ males + 8 females; Delta: n = 4 males + 4 females; Omicron/BA.1: $n = 4$ males + 4 females). **F**–**K** SARS-CoV-2 detection in the brainstem at 4 dpi (F-H) and 80 dpi (**I**–**K**) in male and female hamsters. **F**, **I** Genomic and sub-genomic viral RNA were

assessed based on the RdRp and E gene sequence. Gray lines connect symbols from the same animals. **G**, **J** Isolation of infectious virus from the brainstem. Heatmaps showing positive isolation and amplification (red squares) or negative isolation (gray squares) of infectious virus. Each square corresponds to one different hamster. The number of animals in each experiment is marked in the graphs. **H** Immunofluorescence imaging of SARS-CoV-2 isolation from the brainstem of hamsters at 4 dpi. The micrograph is representative of the positive results shown in (**G**), where the virus was detected in Wuhan ($n = 4$ males + 4 females), Delta ($n = 4$ males + 3 females) and Omicron/BA.1 ($n = 4$ males + 4 females). **K** Immunofluorescence imaging of SARS-CoV-2 isolation from the brainstem of hamsters at 80 dpi. The micrograph is representative of the positive results shown in (J), where the virus was detected in Wuhan ($n = 3$ males + 3 females), Delta ($n = 4$ males + 3 females) and Omicron/BA.1 ($n = 3$ males + 3 females). **H**, **K** Labeling for the SARS-CoV-2 spike (S, green) and nucleoprotein (NP, magenta). The nuclei of Vero-E6 cells are stained with Hoechst (blue). Scale bar: 75 μm. Related to Supplementary Fig. 1-4.

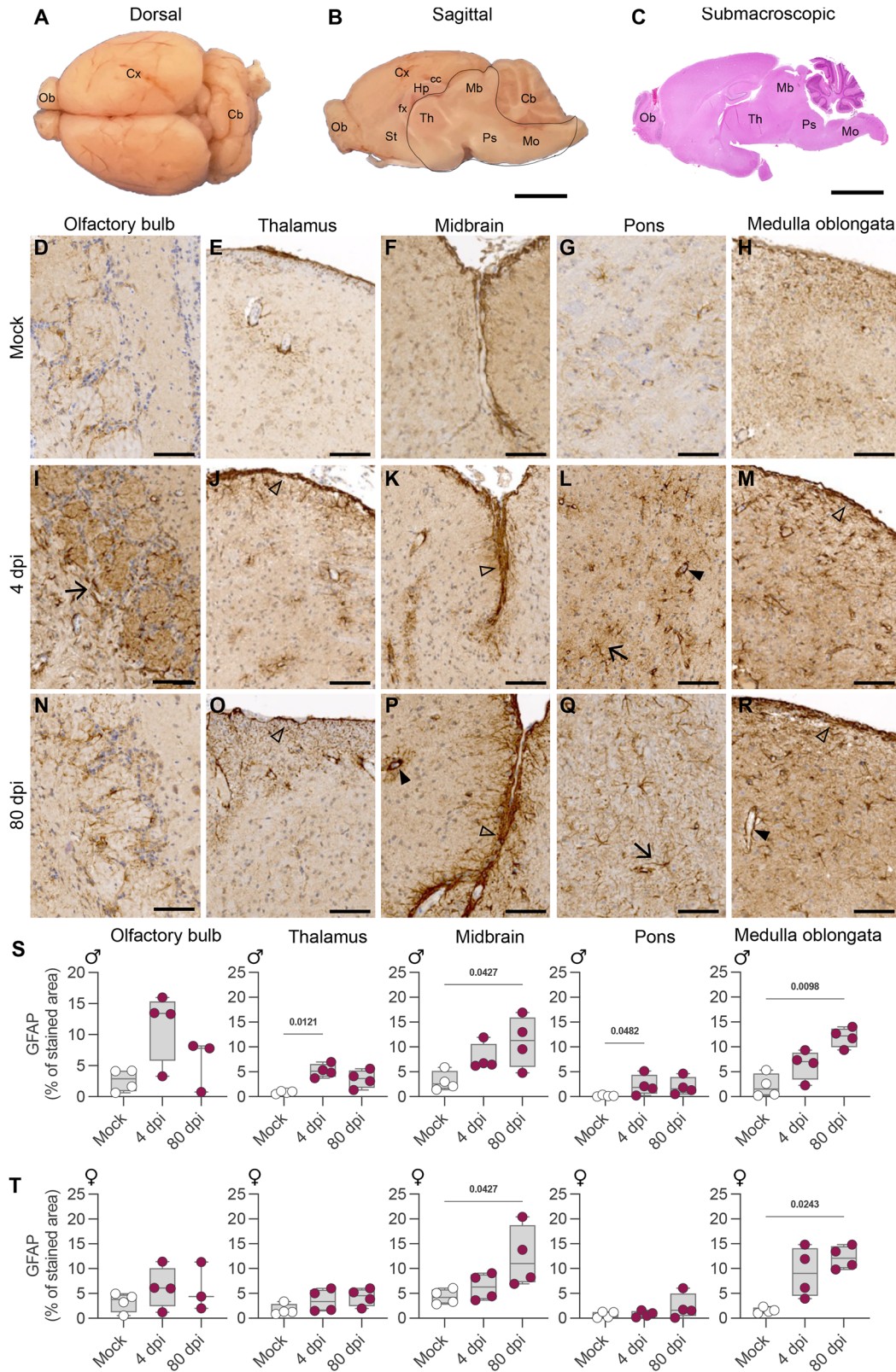

infected males and females at 4 dpi (Supplementary Fig. 5AE–AF). Additionally, mild microgliosis was detected in the thalamus of females (Supplementary Fig. 5V). No distinguishable microglial changes were observed in other regions at 4 dpi (Fig. 2W–Y) or at 80 dpi (Fig. 2Z–AD), except an increase in the IBA-1 detection in the medulla oblongata of infected male hamsters (Supplementary Fig. 5AE).

## The brainstem presents distinct transcriptomic profiles in acute and late infection

Next, willing to unveil if the metabolism of the brainstem was affected during SARS-CoV-2 infection and persistence, we performed a comparative agnostic transcriptomic approach using bulk RNA-seq in the brainstem of hamsters in two distinct comparisons: acute infection (brainstems of male and female infected hamsters at 4 dpi in

**Fig. 2 | Histopathological analysis in the brain of hamsters intranasally-inoculated with SARS-CoV-2 Wuhan. A** Dorsal view of a formalin-fixed hamster brain. Ob: olfactory bulbs, Cx: cortex, Cb: cerebellum. **B** Midsagittal section of a formalin-fixed hamster brain. Hp: hippocampus, cc: corpus calosum, fx: fornix, St: striatum, Th: thalamus, Mb: midbrain, Ps: pons, Mo: medulla oblongata. The brainstem (contoured by a black line) can be separated from the brain after removal of the cerebellum and detachment from the telencephalon at the level of the corpus callosum and the fornix. **C** Submacroscopic view of a sagittal section (HE staining), with indications of the regions from where the images **D**–**R** were obtained. Astrocytes (GFAP + ) detection in the brain of a mock-infected hamster: olfactory bulb (**D**), thalamus (subependymal zone, **E**), midbrain at the level of the cruciform sulcus between the superior and inferior colliculus (**F**), pons (**G**) and medulla oblongata (subependymal zone; **H**). In the olfactory bulbs, astrogliosis is evident only at 4 dpi (**I**, arrow). In the brainstem, however, GFAP-immunoreactive astrocytes are increased in both acute COVID-19 (4dpi, **J**–**M**) and long Covid (80

dpi, **O**–**R**), where the *glia limitans superficialis* (subpial astrocyte foot processes; open arrowhead), the *glia limitans perivascularis* (perivascular astrocyte foot processes; filled arrowhead) and parenchymal astrocytes (arrow) are more evident. **D**–**R**: Representative images of immuno-histochemistry to detect astrocytes using GFAP as marker in mock-infected ($n = 4$ males + 4 females), hamsters infected at 4 dpi ($n = 4$ males + 4 females), and at 80 dpi ($n = 4$ males + 4 females). Scale bars: A-C = 5 mm, D-R = 100 μm. **S, T** Quantification of the percentage of the GFAP stained area in the olfactory bulb, thalamus, midbrain, pons, and medulla oblongata of male (**S**) and female (**T**) hamsters. Mock-infected ($n = 4$ males + 4 females), hamsters infected at 4 dpi ($n = 4$ males + 4 females), and at 80 dpi ($n = 4$ males + 4 females). Box and whisker plots (median, first and third quartiles, minimum and maximum). Individual values are also shown. Kruskal-Wallis test followed by the Dunn's multiple comparisons test (the adjusted $p$ value is indicated if $p < 0.05$). Related to Supplementary Fig. 5.

comparison with age-related mock-infected animals), and late infection (brainstems of male and female infected hamsters at 80 dpi in comparison with age-related mock-infected animals).

In the brainstem at 4 dpi, we observed 3863 and 4014 differentially expressed genes (DEGs; increased or decreased, respectively), from which 391 and 115 DEGs (increased or decreased, respectively) had a fold change higher than 2 (Supplementary Fig. 6). The DEGs were classified according to the GO (gene ontology) terms based on their biological processes, cellular components and molecular functions, and to the KEGG (Kyoto Encyclopedia of Genes and Genomes) pathways. Regarding GO terms, dysregulated biological processes (366 GO terms) were mainly related to inflammation, pyroptosis and cytolysis, type I interferon and viral processes; cellular component (117 GO terms) were related to neurites, synapses and mitochondria, whereas molecular function (55 GO terms) highlighted energy production and channel activity (Fig. 3). Sixty-nine KEGG pathways were significantly regulated, including pathways related to neurodegeneration, neurotransmission, energy, cellular metabolism, autophagy, intracellular signaling, and viral cycle (Fig. 4A).

At 80 dpi, we observed 410 and 424 DEGs (increased or decreased, respectively), from which 7 and 94 DEGs (increased or decreased, respectively) with a fold change higher than 2 in the brainstem (Supplementary Fig. 6). During this late phase of SARS-CoV-2 infection, the dysregulated biological processes in the brainstem (234 GO terms) mainly involved energy, synapses, axonogenesis and neurogenesis; the cellular component (131 terms) was identified mainly as targeting synaptic zones and mitochondria; and molecular function (28 GO terms) was related to energy and calcium activity (Fig. 3). Interestingly, two biological processes terms related to behavior, memory and cognition, were also identified (Fig. 3A). Further, twenty-three KEGG pathways were significantly regulated, mostly related to neurodegeneration, neurotransmission, calcium signaling and energy (Fig. 4A). Remarkably, several GO terms and KEGG pathways were found to be regulated in the brainstem at both 4 dpi and 80 dpi (Figs. 4B–D, 5).

### The synapse system seems to play a central role in the long-term consequences of COVID-19

Among the common dysregulated pathways observed in the brainstem at 4 dpi and 80 dpi, the dopaminergic and glutamatergic synapses pathways can be highlighted (Fig. 4A). Regarding the dopaminergic pathway, on the one hand, there is a significant reduction in the gene expression related to the enzymes involved in dopamine biosynthesis *TH* (tyrosine hydroxylase) and *DDC* (dopa decarboxylase) at 4 dpi, while the genes of dopamine receptors (*DRD1*) and intracellular mediators (*PPP1R1B*, also known as *DARP-32*; and *CAMK2A*) are overexpressed (Fig. 4B). At 80 dpi, on the other hand, there is a decrease of the mRNA expression of dopamine receptors (*DRD1, DRD2*), AMPA receptors (*GRIA1, GRIA2*), and intracellular mediators

(*PPP1R1B, CAMK2A*), with an increase of few genes, including *FOS, TH* and *DDC* (Fig. 4B).

The glutamatergic synapse pathway followed the same profile as the dopaminergic one, with an upregulation of mRNAs related to glutamate receptors from different categories (AMPA, NMDAR, metabotropic, ionotropic) and down-regulation of key enzymes (glutaminase: *GLS2*) and different intracellular mediators (*GNB2, GNB3, GNG3, GNG5, ADCY5, MAPK3*) at 4 dpi. A more restricted number of DEGs was observed at 80 dpi, including glutamate receptors (*GRIK5, GRIA2*), and intracellular mediators, including phospholipases (*PLA2G4E, PLD1*; Fig. 4C).

Further, the synaptic vesicle cycle pathway was also dysregulated, which includes the mRNA expression of key members of the SNARE-complex (Munc18: *STXBP1*; and complexin: *CPLX2*), of the clathrin-mediated endocytosis (clathrin: *CLTA, CLTB, CLTC*; dynamin: *DNM2, DNM3*) and of neurotransmitter transporters (glutamate: *SLC1A2*, also known as *EAAT2; SLC17A6*, also known as *VGLUT2*; GABA: *SLC6A13*, also known as *GAT2*; Fig. 4D). Further, the mRNA expression of other neurotransmitter receptors was also impaired in the brainstem of infected hamsters, including downregulation of *CHRM4* and *HTR2C* at 80 dpi (Supplementary Fig. 7).

### SARS-CoV-2 infection affects the expression of genes related to neurodegenerative processes in the brainstem

Neurodegeneration was also a term frequently observed in the RNA-seq analyses in the brainstem of hamsters infected with SARS-CoV-2. All KEGG pathways from the database "Neurodegenerative disease" were dysregulated in both at 4- and 80-dpi, namely: Alzheimer's disease, Parkinson's disease, Amyotrophic lateral sclerosis, Huntington disease, Spinocerebellar ataxia, Prion disease, Pathways of neurodegeneration - multiple diseases (Fig. 4A). The infection by SARS-CoV-2 dysregulated the expression of a multitude of genes in the brainstem that can contribute, trigger or exacerbate neurodegenerative processes (Fig. 5), including the following seven main processes considered hallmarks of neurodegeneration[25] *i*: altered energy homeostasis (altered mitochondrial electron transport, complexes I to V) which includes some genes common to the oxidative phosphorylation pathway (shown in orange in Fig. 5). *ii*: synaptic and neuronal network defects (dysregulated synaptic vesicle cycle; abnormal dopaminergic and glutamatergic neurotransmission; Fig. 4B–D). *iii*: aberrant proteostasis (altered proteasome activity, shown in green in Fig. 5). *iv*: genomic instability (dysfunctional base excision repair, nucleotide repair, spliceosome), and *v*: inflammation (more evident in the acute phase; Fig. 3A, Supplementary Fig. 8). Our results further include molecular evidence of *vi*: neuronal cell death (loss of trophic support such as BDNF, ErbB signaling) and *vii*: stem cell exhaustion (decreased neurogenesis, axonogenesis).

In a review on the hallmarks of neurodegeneration, Wilson et al.[25] present a list of genes with the highest risk of causing neurodegenerative

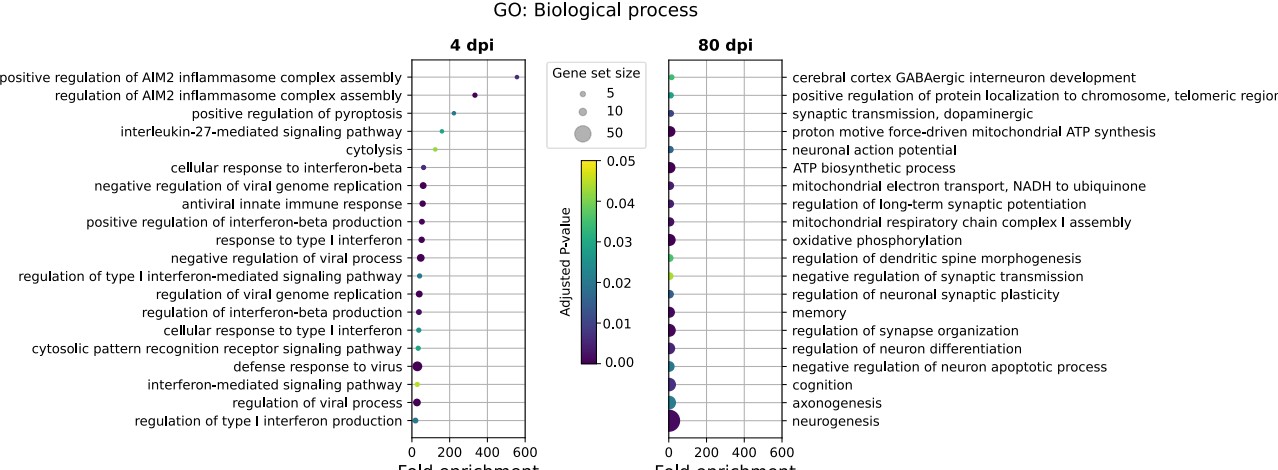

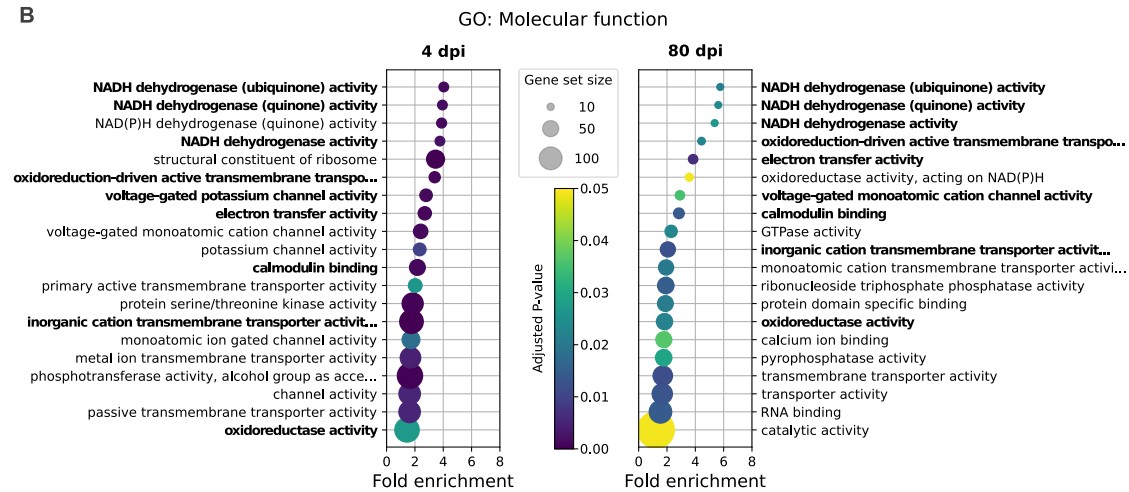

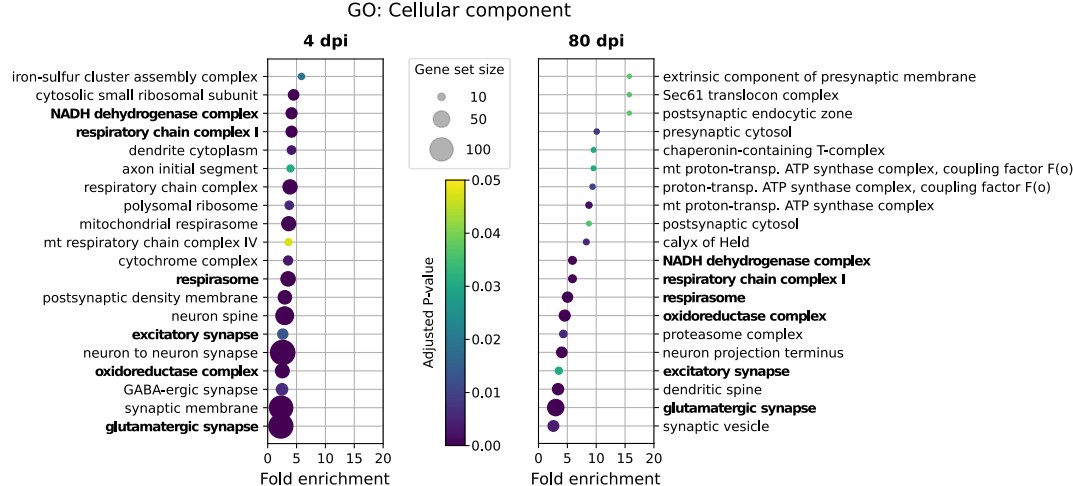

**Fig. 3 | Intranasal SARS-CoV-2 infection alters the brainstem transcriptomic profile in hamsters.** GO enrichment analysis considering biological process (**A**), molecular function (**B**) and cellular compartment (**C**). Selected GO terms are based on the up- and down-regulated genes between SARS-CoV-2 Wuhan-infected and mock-infected samples at 4 dpi (left panels) and 80 dpi (right panels). Circle sizes are proportional to the gene set size, which shows the total size of the gene set associated with the GO terms. Circle color is proportional to the corrected *p* values (one-sided tests, not considering potential depletions, and corrected for multiple testing using Benjamini-Hochberg correction). Common pathways observed at 4 and 80 dpi are shown in bold. Related to Supplementary Figs. 6–8.

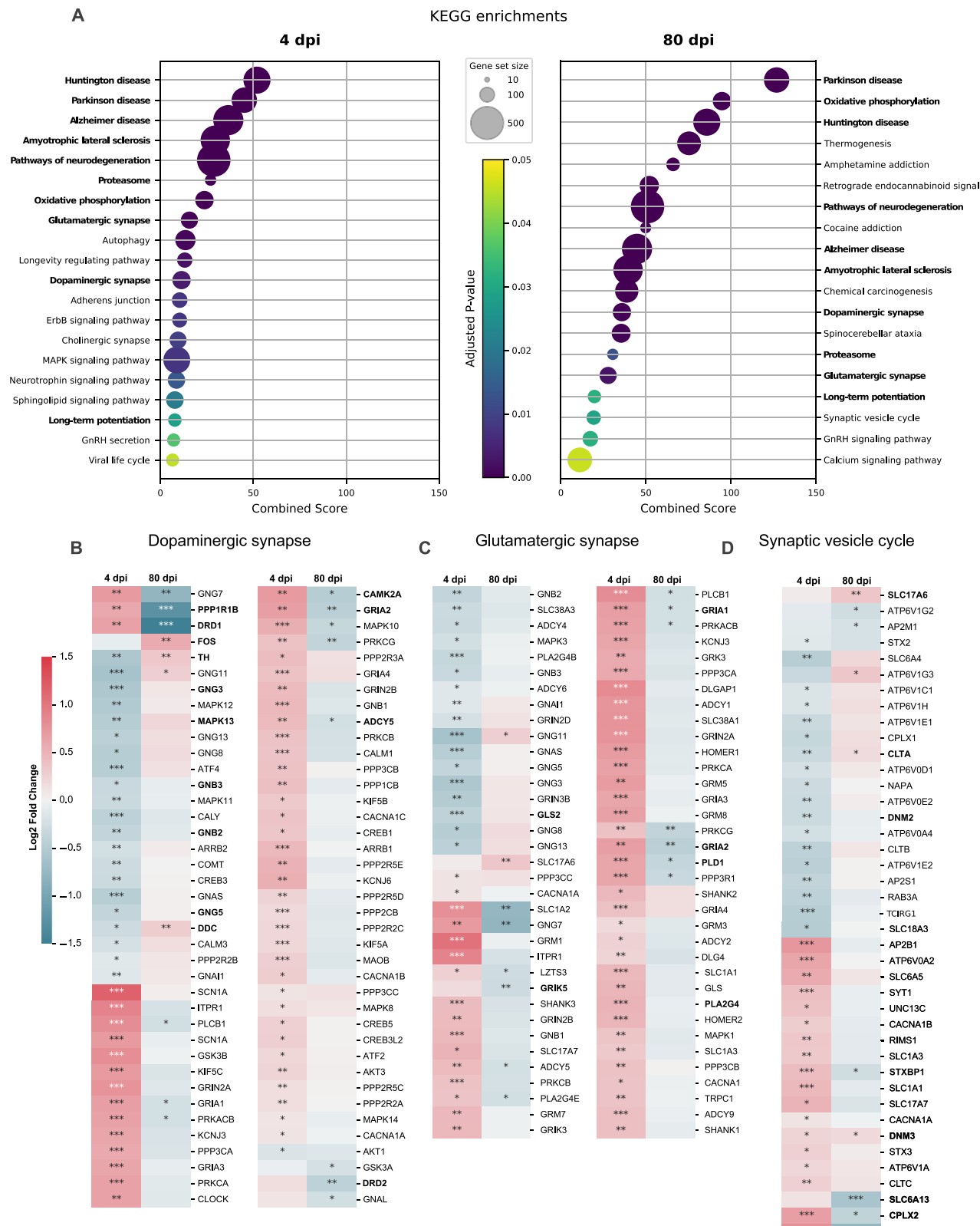

**A** KEGG enrichments

**B** Dopaminergic synapse

**C** Glutamatergic synapse

**D** Synaptic vesicle cycle

diseases in humans. Among them, we found that the expression of *ATXN1, HTT, KIF5A, LRRK2 (*also known as *PARK8), MAPT, PARK7 (*also known as *DJ-1), PSEN2, SQSTM1, SOD1,* and *TBK1)* was dysregulated in the brainstem of the infected hamsters at 4 dpi. In addition, the expression of *APP, LRRK2, PARK7, PDYN, SOD1,* and *VCP* was dysregulated in the brainstem at 80 dpi (Fig. 5).

Lastly, we evaluated the gene expression of selected targets by RT-qPCR in the brainstem of male and female hamsters infected with Wuhan, Delta, and Omicron/BA.1 at 4 and 80 dpi to corroborate the involvement of the abovementioned signaling pathways identified by RNA-seq. Regarding dopaminergic and glutamatergic synapses impairment, the genes *DRD2, FOS, CAMK2A, PLA2G4E, CHRM4, GRID1*

**Fig. 4 | Intranasal SARS-CoV-2 infection alters the brainstem transcriptomic profile in hamsters.** **A** KEGG pathways enrichment based on the differentially regulated genes between SARS-CoV-2 Wuhan-infected and mock-infected samples at the acute phase (4 days post-infection, dpi; left panel) and at the late phase (80 dpi, right panel). Circle sizes are proportional to the gene set size. Circle color is proportional to the corrected *p* values (one-sided tests, not considering potential depletions, and corrected for multiple testing using Benjamini-Hochberg correction). Common pathways observed at 4 and 80 dpi are shown in bold. Heatmap

showing the differentially expressed genes at 4 and 80 dpi according to the selected KEGG pathway "Dopaminergic synapse" (**B**), "Glutamatergic synapse" (**C**), and "Synaptic vesicle cycle" (**D**), calculated in comparison with mock-infected hamstersColor gradient represents the transcription log2 fold change comparing infected and mock-infected (two-sided test, adjusted p-values after Benjamini-Hochberg multiple testing correction, considering a false discovery rate FDR < 0.05: *, **, *** denote *p* < 0.05, *p* < 0.01 and *p* < 0.001, respectively). Gene names in bold are mentioned in the text. Related to Supplementary Fig. 6-8.

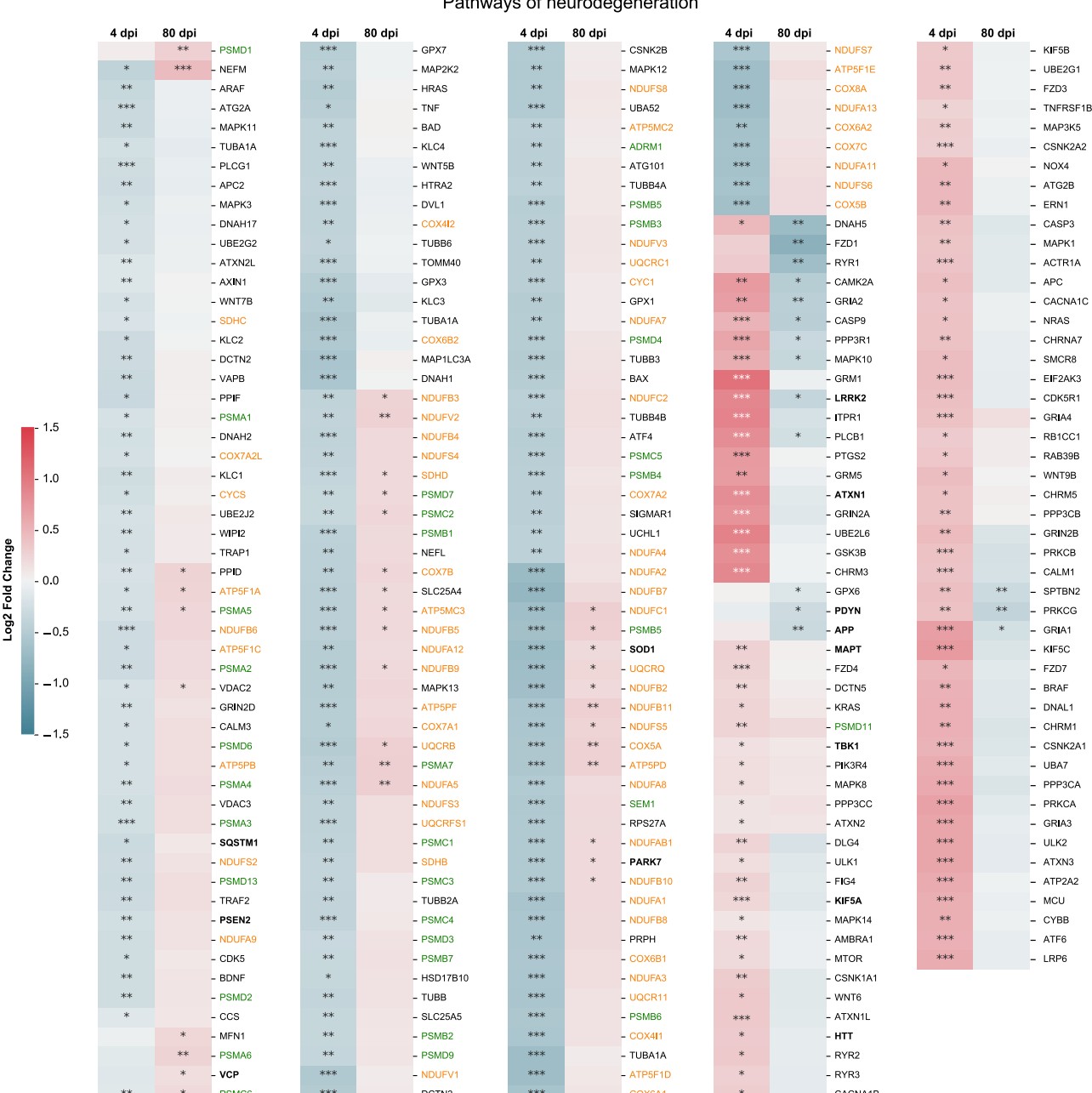

Pathways of neurodegeneration

**Fig. 5 | Intranasal SARS-CoV-2 infection affects the expression of genes related to neurodegenerative processes in the brainstem.** Heatmap showing the differentially expressed genes at 4- and 80-days post-infection (dpi) according to the selected KEGG pathway "Pathways of neurodegeneration" calculated comparing SARS-CoV-2 Wuhan-infected to mock-infected hamsters (two-sided test, adjusted p-values after Benjamini-Hochberg multiple testing correction, considering a false

discovery rate FDR < 0.05: *, **, *** denote *p* < 0.05, *p* < 0.01 and *p* < 0.001 respectively). Common genes appearing in the 'Oxidative phosphorylation pathway' are shown in orange, whereas common genes appearing in the 'Proteasome pathway' are shown in green. The color gradient represents the transcription log2 fold change comparing infected and mock-infected. Gene names in bold are mentioned in the text. Related to Supplementary Fig. 6-8.

and *SOX2* were found upregulated in the brainstem of males and females at 4 dpi, whereas *DRD1* and *ADCY5* were upregulated only in females. Differences in the gene expression were less intense at 80 dpi, nevertheless, we observed a downregulation of *HTR2C* in the brainstem of both males and females; downregulation of *DRD1, PLA2G4*, and *GRID1* in females; and downregulation of *DRD2* and *CAMK2A* in males (Supplementary Fig. 7).

The genes *MX2, ISG20, IFNL* and *CXCL10* were upregulated in the brainstem of both male and female hamsters at 4 dpi, in agreement with the pathways related to inflammation and antiviral innate immune response in the acute phase (Supplementary Fig. 8). Concerning neurogedeneration, oxidative phosphorylation, and proteasome pathways at 4 dpi, *PSMB5* expression was reduced in male brainstems; *CHRM1* and *ATP5ME* were upregulated in female brainstems. In contrast, at 80 dpi, *PSMB5* was upregulated in the brainstem of both sexes, whereas in males, *COX5A* and *ATP5ME* were upregulated and *CHRM1* was downregulated (Supplementary Fig. 8).

### SARS-CoV-2-infected hamsters manifest long-term neuropsychiatric symptoms and cognitive deficit

After demonstrating that SARS-CoV-2 infects and persists in the brainstem, and that the brainstem of infected animals presents a neurodegenerative molecular signature, we decided to test whether infected animals also exhibit clinical manifestations related to long Covid, specifically anxiety, depression, and memory loss, which may be related to neurodegenerative processes. To this end, we performed three well-described and recognized behavioral tests to assess anxiety, depression, and recognition memory in the animals. The hamsters performed four sequential behavioral tests (one test/day) on three sessions: the first session just after the acute phase (between 14-17 dpi, hereinafter called 15 dpi session), the second between 28-31 dpi (hereinafter called 30 dpi session), and the third session between 76-79 dpi (hereinafter called 80 dpi session) (Fig. 6A). To properly quantify the effect of covariates (sex, time post-infection, SARS-CoV-2 variant), we used mixed model regression to account for within-individual correlation arising from repeated measurements.

First, we evaluated signs of anxiety by using the well-known light-dark box and the novelty-suppressed feeding tests. During the novelty suppressed feeding test, infected animals displayed anxiety-like behavior, referred to as hyponeophagia (i.e., inhibition of feeding caused by a new environment: anxiety behavior = increased latency to eat) at the 80 dpi session (Fig. 6B, E). This phenomenon was more pronounced in male hamsters infected with SARS-CoV-2 Wuhan, and to a lesser extent with Omicron/BA.1 (Supplementary Fig. 9). Regarding the light-dark box (which is based on rodents' innate aversion to light), this test was unable to differentiate anxious behavior among the groups (Supplementary Fig. 10), which may have been affected by different factors, including the robustness of the test and the lighting conditions of the BSL-3 isolators room.

Next, we assessed depressive-like behavior by carrying out the sucrose splash-test. This test consists of comparing the time that the animals spend grooming after spraying a 10% sucrose solution on their backs: the time grooming corresponds to an index of motivational and self-care behavior (decreased time grooming indicates depression-like behavior). The test showed a dichotomy between males and females: Wuhan-infected females exhibited persistent depressive-like behavior (15, 30, and 80 dpi sessions), whereas Delta-infected females presented a relapsing depressive-like behavior (15 and 80 dpi sessions). Omicron/BA.1-infected females, however, exhibited no evidence of a depressive-like behavior (Fig. 6C, F, Supplementary Fig. 11). No depressive-like behavior was noticed in infected male hamsters.

Lastly, to assess the short-term memory performance of the infected animals, we used the novel object recognition test, where a low discrimination index indicates impaired recognition memory. Our statistical model revealed that both male and female animals infected

with Wuhan exhibited memory impairment over time (Fig. 6D, G), despite the presence of individual variation. In contrast, hamsters infected with the Delta and Omicron/BA-1 variants demonstrated more variable results (Supplementary Fig. 12).

## Discussion

Long Covid is a complex condition that can affect different organs and can be manifested as a multitude of symptoms after COVID-19. Since the first reports by patients, there have been different definitions of long COVID, and the most recent one was proposed by the World Health Organization in February 2025 as "a condition characterized by a range of symptoms that usually start within 3 months of the initial COVID-19 illness and last for at least 2 months. These symptoms may persist from the initial illness or develop after recovery. It can affect a person's ability to perform daily activities such as work or household chores and restrict social participation"[26]. However, despite a well-accepted definition and the existence of different hypothesis and evidence[27–29], the exact underlying mechanisms of long COVID are not yet fully understood, especially those related to the manifestation of neurologic and neuropsychiatric symptoms. Here we describe clinical, behavioral, virologic, and transcriptomic data in the golden hamster, a relevant model to study long COVID. We provide conclusive evidence that neuropsychiatric symptoms and cognitive impairment related to long COVID follow the acute infection, in a model without social or somatization influence, and no effects related to post-intensive care syndrome.

The hamster model presented herein recapitulates long COVID in humans. In human patients living with long Covid, the incidence of cognitive and neuropsychiatric symptoms may vary according to different studies[30–32]: in a meta-analysis regrouping 10,530 patients from 18 studies, 32% of them had brain fog and 28% had memory impairment[33]. Another study reports that among 9605 patients living with long COVID, 22% had anxiety and 21% manifested signs of depression[34]. Remarkably, a recent study with human volunteers that were infected by the original SARS-CoV-2 Wuhan strain showed that the infection led to a persistent deficit in recognition memory and spatial planning, even if the infected volunteers did not report subjective cognitive deficit[15].

Although behavioral testing usually requires large numbers of animals to achieve statistical significance, our long-term, sequential behavioral testing approach, coupled with mixed-model regression analysis, revealed that infected hamsters also exhibit persistent depression-like behavior, impaired recognition memory, and delayed signs of anxiety after acute infection, which can persist or fluctuate over time. These cognitive and neuropsychiatric symptoms can have different origins and causes; though, the asset of the hamster model is that the behavioral tests took place in an environment free from social pressure or risk of somatization, i.e., infected and mock-infected animals (both male and female) were tested simultaneously under the exact same conditions. Consequently, the differences observed in clinical profile between the infected and mock-infected animals can only be explained by the SARS-CoV-2 infection itself, and the underlying mechanisms are likely to be associated with virus-related and neuro-immunometabolic changes in their brainstem.

Virus-related mechanisms include SARS-CoV-2 persistence, either infectious (replicative) viral particles or viral components in the brain. The neuroinvasive nature of SARS-CoV-2 is a key point regarding the neurological sequelae associated with long COVID[35]. It has already been demonstrated in human COVID-19 post-mortem samples[17,36–40] and reproduced in animal models[8,9,41–44] that SARS-CoV-2 infects the brain, despite some other studies not detecting the infection[45,46]. Different factors, such as time-points post-infection, patient's clinical history, and detection methods, can affect SARS-CoV-2 detection in the brain. Currently, we cannot address the question of SARS-CoV-2 brain infection and viral replication in living patients because there are

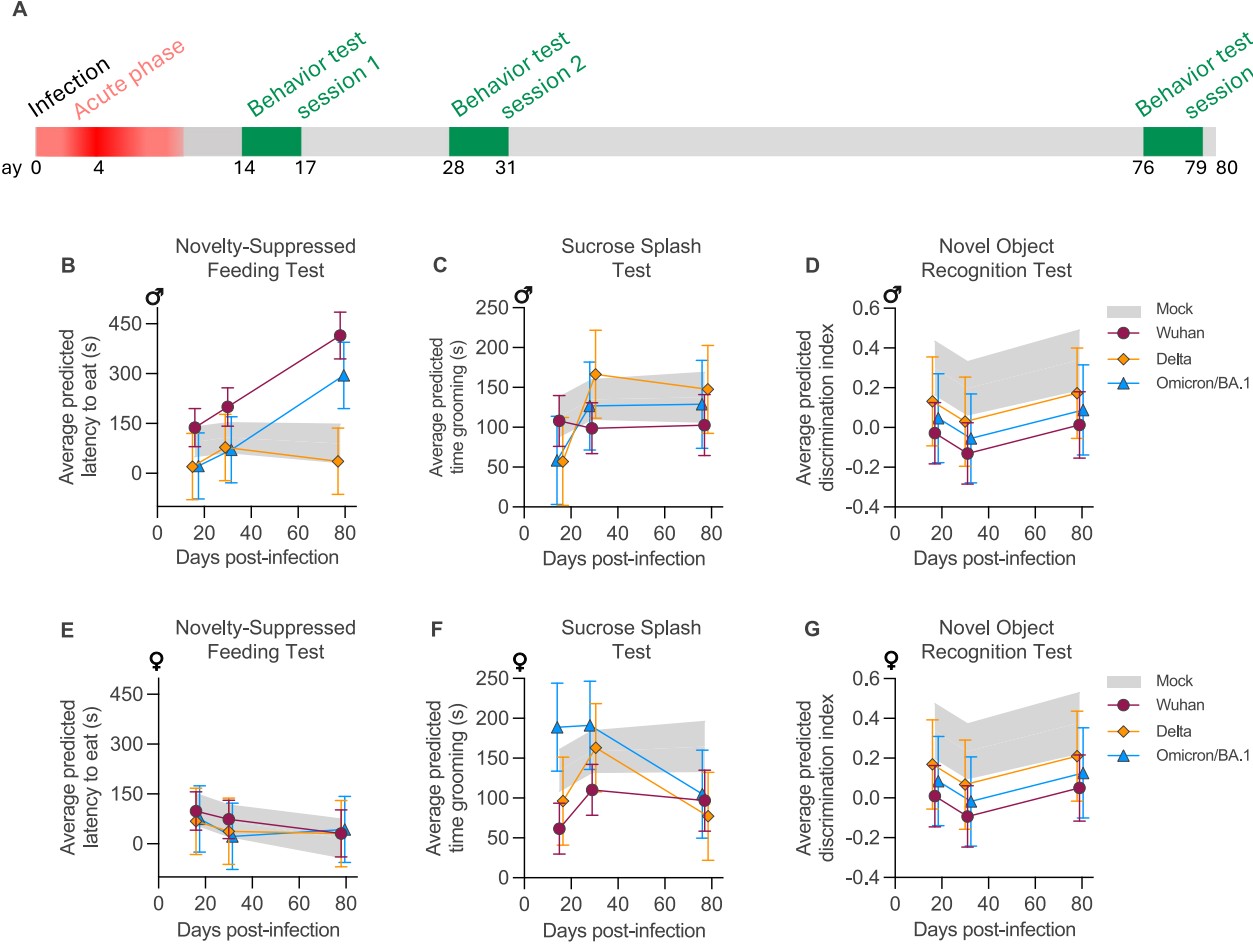

**Fig. 6 | Long-term impact of SARS-CoV-2 infection on neuropsychiatric and cognitive behavior in hamsters. A** General experimental outline with the three sessions of behavior testing. Novelty-suppressed feeding test: Average predicted latency to eat analyzed by mixed model regression in male (**B**) and female (**E**) hamsters. Increased latency to eat indicates hyponeophagia and corresponds to an anxious behavior. Sucrose splash test: Average predicted time grooming analyzed by mixed model regression in male (**C**) and female (**F**) hamsters. Time grooming corresponds to an index of motivational and self-care behavior and decreased grooming time is considered as a depression-like behavior. Novel object recognition test: Average predicted discrimination index analyzed by mixed model regression in male (**D**) and female (**G**) hamsters. Decreased discrimination index corresponds to less time spent exploring the new object and is indicative of short-term memory impairment. Data were analyzed using a mixed-model regression based on time post-infection (Day), Sex, and viral variant (Infection), with a random effect accounting for within-individual variability. For the novelty-suppressed

feeding test and the sucrose splash test, the model used was Day * Sex * Infection. For the novel object recognition test, the model used was Day + Sex + Infection. For each outcome, model selection was performed to preserve the most parsimonius parameterization. Complete analyses are available in the Git repository [https://gitlab.pasteur.fr/hub/19421-long-covid-brainstem]. Horizontal lines indicate the estimated means and the 95% confidence interval. The gray crosshatched zone indicates the estimated means and the 95% confidence interval of the mock-infected group. Behavioral session 1 (mock-infected: $n = 20$ males + 18 females; Wuhan: $n = 12$ males + 12 females; Delta: $n = 4$ males + 4 females; Omicron/BA.1: $n = 4$ males + 4 females). Behavioral session 2 (mock-infected: $n = 20$ males + 18 females; Wuhan: $n = 12$ males + 12 females; Delta: $n = 4$ males + 4 females; Omicron/BA.1: $n = 4$ males + 4 females). Behavioral session 3 (mock-infected: $n = 12$ males + 12 females; Wuhan: $n = 8$ males + 8 females; Delta: $n = 4$ males + 4 females; Omicron/BA.1: $n = 4$ males + 4 females). Related to Supplementary Figs. 9–12.

no biomarkers to assess active infection and viral persistence in the CNS. This is a major advantage of the hamster model, which mimics the human disease and may help to elucidate this important aspect of the pathogenesis of long COVID.

As previously reported in experimental infections with SARS-CoV[47] and MERS-CoV[48], we show herein that SARS-CoV-2 quickly reaches the brainstem after intranasal inoculation. We also show thet there is an active replication of all tested variants (Wuhan, Delta, Omicron/BA.1) during the acute phase of the infection, which corroborates other studies reporting brainstem infection in human cases of acute COVID-19[37–39]. Surprisingly, we report the unprecedented isolation of infectious viral particles from the brainstem of hamsters with long COVID 80 days after infection, regardless of sex and SARS-CoV-2 variant. This contradicts the common assumption that there is no infectious virus in the brain[17,49]. It also highlights the need to use later

time points than just 30 days when working with small animals to reproduce long COVID. However, the infectious virus titer in the hamster brainstem was quite low in both acute (4 dpi) and late phases (80 dpi) of the infection, which may contribute to the difficulty in detecting infectious virus in the post-acute phase in some previous studies[17,41]. We could not define whether the virus was restricted to specific areas or in specific cell populations, but we demonstrate that the brainstem may act as a reservoir of infectious virus during the post-acute phases of COVID-19[20]. Concomitant to virus-related mechanisms, we found significant neuro-immunometabolic changes in the brainstem of the infected hamsters, which can be divided into three major categories: inflammation, neurotransmission, and energy. This agrees with a previous transcriptomic study that reveals four equivalent impaired processes in different brain regions of hamsters at 3 dpi and 30 dpi: metabolism, synaptic signaling, neural plasticity, and

immune response[41]. Signs of inflammation in the brainstem of infected hamsters were better observed at 4 dpi, where an antiviral response characterized by the expression of IFN-related genes (*ISG15, MX2, IRF7*) occurs, along with microgliosis in the thalamus. Such an immune profile recapitulates what has been demonstrated in humans[11,36] and in other brain regions of hamsters[9] during the acute phase. However, inflammation may last longer[11,50,51].

Contrastingly, neurotransmission seems to be greatly affected during both acute and long COVID, and infected hamsters presented a transcriptomic profile highly indicative of dopaminergic and glutamatergic impairment. We show a persistent alteration in dopaminergic signaling with dysregulation in the gene expression of key enzymes related to dopamine synthesis (tyrosine hydroxylase, DOPA decarboxylase) and dopamine receptors (*DRD1, DRD2*). Most of the dopaminergic neurons in the mammal brain reside in the midbrain (part of the brainstem), are the main source of dopamine[52], and play an essential role in behavior and the reward system[53]. Previous studies have shown that dopaminergic neurons in the midbrain were found positive for SARS-CoV-2[39] and that the density of dopaminergic neurons was reduced in patients who died of acute COVID-19[54]. The susceptibility of dopaminergic neurons to the infection has been linked to an inflammatory response and cellular senescence[54]. In addition, the infection has been shown to have the potential to exacerbate Parkinson's disease by inducing the death of dopaminergic neurons[55]. With the hamster model described here, we provide additional evidence that the disruption of the dopaminergic network may persist for a long time after the acute phase of the disease.

Parallelly to dopamine, impaired glutamate signaling may also play a role in mental fatigue, mood disorders, and cognition deficits[56–58], and it was recently described that some dopaminergic neurons in the midbrain can also co-transmit glutamate[59,60]. Glutamate is the main excitatory neurotransmitter in the brain[61] and it's involved in learning and memorization[62]. Besides neurons, astrocyte dysfunction may further impair glutamatergic signaling and be involved in the pathophysiology of long Covid[63]; astrocytes are responsible for most glutamate removal from the synaptic cleft via the receptor *SLC1A2* (also known as glutamate transporter 1 - *GLT1* or excitatory amino acid transporter 2 -*EAAT2*)[57], found down-regulated in the brainstems of the hamsters with long Covid. Finally, we cannot exclude the involvement of other neurotransmission pathways, as GABAergic and cholinergic dysregulation have equally been reported during SARS-CoV-2 infection[64,65]. In agreement with other studies, the results presented herein also advocate for a broader impact on synapses, either via dysregulation of synaptic members gene expression[21], destruction or degeneration of synaptic circuits[66], and even physical obstruction to the neurotransmission due to changes in synapse morphology[21], to viral-induced neuronal fusion[67], neuroaxonal damage[68], or even loss of myelin[69].

Along with impaired neurotransmission, energy metabolism, and mitochondrial dysfunction may also be involved in neuropsychiatric pathologies, including depression[70]. Intriguingly, zones of hypometabolism in various brain areas, including the brainstem, have been identified by [18F]fluorodeoxyglucose (FDG)-PET in patients living with long COVID[71–74]. Additionally, a magnetic resonance imaging study including 401 patients with a positive diagnosis of SARS-CoV-2 infection revealed a significant difference in the volume of the brainstem, besides tissue damage in areas functionally connected to the primary olfactory cortex[75]. In order to maintain synaptic activity, mitochondrial function and energy supply must be tightly controlled, and deregulation of these processes can result in neurodegenerative damage[25].

Altogether, the structural, metabolic, and synaptic changes observed in our data are also hallmarks of neurodegenerative diseases, including the expression of genes highly correlated to neurodegeneration in humans[25]. The upregulation of innate immunity genes and dowregulation of energy metabolism and proteostasis genes have

been appointed as a pan-neurodegenerative gene signature, implying neuronal function loss[76]. Altogether, the results presented herein argues for a neurodegenerative molecular signature in the brainstem induced by SARS-CoV-2 infection, a characteristic shared by some human neurodegenerative diseases[77,78] and which may explain the long-term symptoms observed during long Covid[50,51,79]. We do not advocate that SARS-CoV-2 causes neurodegenerative diseases, instead, our findings highlight that the infection disrupts important metabolic pathways, already in the acute phase, which can trigger neurodegenerative processes and cause neurobehavioral changes, as may be the case in other viral encephalitis[80–82].

Health agencies and patients' associations alert that patients' access to appropriate care is not always well organized, that health professionals are not well informed, and that there may be a tendency to "stigmatize" and "psychiatrize" symptoms, with health professionals quickly categorizing long Covid symptoms as somatoform[83,84]. In this study, we demonstrate that golden hamsters develop persistent and late-onset neurological symptoms following SARS-CoV-2 infection in a model without social or somatization influence. We describe hamsters as a robust animal model of long COVID and provide evidence that viral and neuro-immunometabolic mechanisms coexist in the brainstem, which exhibits a neurodegenerative signature characterized by an elevated innate immune response and impaired neurotransmission and energy metabolism. This model also illustrates the complexity of long COVID with clinical manifestations varying according to time, sex, and viral variants. Finally, this study highlights that SARS-CoV-2 has the potential to persist in the brain and that the brainstem may act as a reservoir of infectious virus during the post-acute phases of COVID-19. All these findings should support a reinforcement of studies investigating the causes of long COVID directly in humans or in valuable and relevant surrogate models such as the hamster described here.

## Methods
### Ethics
All animal experiments were performed according to the French legislation and in compliance with the European Communities Council Directives (2010/63/UE, French Law 2013–118, February 6, 2013) and according to the regulations of the Pasteur Institute Animal Care Committees. The Animal Experimentation Ethics Committee (CETEA 89) of the Institut Pasteur approved this study (200023; APAFIS#25326-2020050617114340 v2) before experiments were initiated.

### SARS-CoV-2 virus and variants
The SARS-CoV-2 Wuhan BetaCoV/France/IDF00372/2020 isolate (EVAg collection, Ref: 014V-03890) was obtained from the National Centre for Respiratory Viruses (Paris, France). The SARS-CoV-2 Delta/2021/I7.2 200 isolate (GISAID ID: EPI_ISL_2029113) and the SARS-CoV-2 Omicron/B1.1.529 isolate (Omicron/BA.1, GISAID ID: EPI_ISL_6794907) were given by the Virus and Immunity Unit of the Institut Pasteur.

Viral stocks were produced on Vero-E6 cells at an MOI (Multiplicity of Infection) of $10^{-4}$. After three days of infection, the supernatant was collected, clarified, and aliquoted for long-term storage at −80 °C. Viral stocks were titrated on Vero-E6 by classical plaque assays using semisolid overlays (Avicel, RC581-NFDR080I)[85].

### Animal study design and infection
Male and female golden hamsters (*Mesocricetus auratus;* RjHan:AURA) of 5-6 weeks of age (average weight 60-80 grams) were purchased from Janvier Laboratories (Le Genest-Saint-Isle, France) and handled under specific pathogen-free conditions. The animals were housed by groups of 4 and manipulated in isolators in a biosafety level-3 facility in the Institut Pasteur's animal facilities accredited by the French Ministry of Agriculture for performing experiments on live rodents. The animal facility had 14/10 light cycle regimen: 14 hours of light: 10 hour of dark. The animals had *ad libitum* access to water and food. Shredded

cardboard and a plastic ball were placed on the bedding of each cage in the spirit of promoting refinement. Before any manipulation, animals underwent an acclimation period of one week.

Following the 3Rs principle in animal experimentation, some samples used in this study are originated from animals that also took part in other studies, where clinical status and respiratory viral load were already published[8,86].

Animals were anesthetized with an intraperitoneal injection of 200 mg/kg ketamine (Kétamine 1000, Virbac France) and 10 mg/kg xylazine (Rompun, Bayer), and 100 µL of physiological solution containing $6\times10^4$ PFU of SARS-CoV-2 was administered intranasally to each animal (50 µL/nostril). Mock-infected animals received the physiological solution only. Infected and mock-infected hamsters were housed in separate isolators and were followed-up daily which the body weight and the clinical score were noted. The clinical score was based on a cumulative 0-4 scale: ruffled fur, slow movements, apathy, absence of exploration activity.

## Tissue sampling

At 4 hours, 1-, 2-, 4-, 14-, 30- and 80-days post-infection, the animals were euthanized with an excess of anesthetics (ketamine and xylazine) and exsanguination[87]. The brain was extracted from the skull, the two brain hemispheres were separated by a median incision and macroscopically divided in four regions using tweezers: (1) olfactory bulbs, (2) cerebellum, (3) cerebral cortex (containing the cortex, the striatum and the hippocampus) and (4) brainstem (containing the diencephalon, the midbrain, the pons and the medulla oblongata) (Fig. 2). The lungs were extracted from the thorax and weighted. The nasal turbinates were extracted by opening of the nasal cavity after incision of the nasal and frontal bones. The samples were immediately frozen at −80 °C.

Frozen samples were weighed and transferred to Lysing Matrix M 2 mL tubes (116923050-CF, MP Biomedicals) containing 1 mL of ice-cold DMEM (Dulbecco's Modified Eagle medium, Gibco) supplemented with 1% penicillin/streptomycin (15140148, Thermo Fisher). Samples were homogenized using the FastPrep-24™ system (MP Biomedicals), and the following scheme: homogenization at 4.0 m/s for 20 sec, incubation at 4 °C for 2 min, and further homogenization at 4.0 m/s for 20 sec. The tubes were centrifuged at $10,000 \times g$ during 2 min at 4 °C, and the supernatants collected and stored at −80 °C until further analysis.

## Serum sampling and neurofilament light chain (NfL) dosing

At 80-days post-infection, male ($n = 4$) and female ($n = 4$) hamsters infected with SARS-CoV-2/Wuhan, and age-related male ($n = 4$) and female ($n = 4$) mock-infected hamsters were anesthetized (ketamine and xylazine) and blood was collected by intracardiac punction before euthanasia. The blood was store in 1.5 mL tubes and allowed to coagulate at room temperature during at least 30 minutes. The tubes were centrifuged at $2000 \times g$ during 10 min at 4 °C, and the serum was collected and stored at −80 °C. Before NfL dosing, the serum samples (45 µL) were treated with 1% (v/v) Triton X-100 (5 µL) at room temperature for 2 h. The serum levels of NfL were measured using the commercially available single molecule array (SIMOA) assay NF-Light v2 Advantage kit (Quanterix, 104073) on an HD-X analyzer following the manufacturer's instructions.

## RNA extraction

Total RNA from lungs, cerebral cortex, cerebellum and brainstem was extracted using the Direct-zol RNA MiniPrep kit (R2052, Zymo Research). Total RNA from nasal turbinates and olfactory bulbs was extracted using the Direct-zol RNA MicroPrep kit (R2062, Zymo Research). In both cases, 125 µL of tissue homogenate was incubated with 375 µL of Trizol LS (10296028, Invitrogen) and the extraction was performed according to the manufacturer's instructions.

## SARS-CoV-2 detection in golden hamsters' tissues

The detection of genomic and sub-genomic SARS-CoV-2 RNA was based on the RdRP and the E genes[88]. We used the SuperScript III Platinum One-Step qRT-PCR kit (Invitrogen 11732-020) in a final volume of 12.5 µL per reaction in 384-wells PCR plates using a thermocycler (QuantStudio 6 Flex, Applied Biosystems). Briefly, 2.5 µL of RNA was added to 10 µL of a master mix containing 6.25 µL of 2X Reaction mix, 0.2 µL of MgSO4 (50 mM), 0.5 µL of Superscript III RT/Platinum Taq Mix (2 UI/µL), 0.025 µL of ROX Reference and 3.025 µL of nuclease-free water containing 400 nM of primers and 200 nM of probe. To detect the genomic RNA, we used the IP2_primers and probe (IP2_FW 5′-ATGAGCT-TAGTCCTGTTG-3′; IP2_RV 5′-CTCCCTTTGTTGTGTTGT−3′; IP2_Probe FAM-AGATGTCTTGTGCTGCCGGTA-TAMRA, and the E_sarbeco primers and probe (E_Sarbeco_F1 5′-ACAGGTACGTTAATAGTTAATAGCGT-3′; E_Sarbeco_R2 5′-ATATTGCAGCAGTACGCACACA-3′; E_Sarbeco_Probe FAM-5′-ACACTAGCCATCCTTACTGCGCTTCG-3′-TAMRA). The detection of sub-genomic SARS-CoV-2 RNA was achieved by replacing the E_Sarbeco_F1 primer by the CoV2sgLead primer (CoV2sgLead-Fw 5′-CGATCTCTTGTAGATCTGTTCTC-3′). A synthetic gene encoding the PCR target sequences was ordered from Thermo Fisher Scientific. A PCR product was amplified using Phusion™ High-Fidelity DNA Polymerase (Thermo Fisher Scientific) and in vitro transcribed by means of the Ribomax T7 kit (Promega). RNA was quantified using the Qubit RNA HS Assay kit (Thermo Fisher Scientific), normalized, and used as a standard to quantify RNA absolute copy number. The amplification conditions were as follows: 55 °C for 20 min, 95 °C for 3 minutes, 50 cycles of 95 °C for 17 s and 58 °C for 30 s; followed by 40 °C for 30 s.

## Titration, isolation and amplification of SARS-CoV-2 from the brainstem

Initially, we used the classical TCID50 method on Vero-E6 cells to quantify the infectious virus load in the brainstem[89]. Briefly, serial dilutions of homogenized organs (1:5 up to 1:640) were made in DMEM (Dulbecco's Modified Eagle medium, Gibco) supplemented with 1% Penicillin/Streptomycin (15140148, Thermo Fisher) and 1 µg/mL TPCK. We added 100 µL of each dilution to 6 wells of a 96-well plate. Next, 100 µL of a preparation containing $8 \times 10^5$ Vero-E6 per mL of DMEM supplemented with 1% Penicillin/Streptomycin and 1 µg of TPCK is added. The plate was placed in the incubator at 37 °C, 5% $CO_2$ for 2–7 days. The quantification limit is $10^2$ TCID50/mL.

Wells presenting cell lysis and/or cytopathic effect were stained by immunofluorescence. Cell lysis and/or cytopathic effects were observed between 2 and 4 days after exposure for the samples at 4 dpi, and between 5 and 7 days after exposure for the samples at 80 dpi. Briefly, the cells were fixed with 4% paraformaldehyde (15444459, Thermo Scientific), washed in PBS, permeabilised with 0.5% (v/v) Triton 100X for 15 min, washed once in PBS, followed by one hour blocking with 5% (v/v) normal goat serum (10000C, Invitrogen). The primary antibodies mouse anti-nucleocapsid (MA-29981, Invitrogen; dilution 1:500) and rabbit anti-spike (GTX135356, GeneTex; dilution 1:1000), were incubated overnight at 4 °C. After washing with PBS, the secondary antibodies anti-Mouse AF647 (A21235, Invitrogen; dilution 1:1000) and anti-Rabbit AF488 (A11034, Invitrogen; dilution 1:1000) were incubated for 2 hours at RT. The cells were then washed, the nuclei were stained with 20 µM Hoechst 33342 (62249, Thermo Scientific) and stored in PBS.

Further, in an attempt to amplify the virus, the supernatant from wells presenting cell lysis and/or cytopathic effect was recovered, diluted (1:10) and 1 mL/well was added in a 6-well plate containing $1\times10^6$ Vero-TMPRSS2 cells. After one hour, the supernatant was removed and replaced by 2 mL of DMEM supplemented with 1% Penicillin/Streptomycin and incubated during 3 or 7 days, for brainstem samples collected at 4 or 80 dpi, respectively. The cells were then lysed with Trizol LS (10296028, Invitrogen), the RNA was extracted with the Direct-zol RNA MicroPrep kit (R2062, Zymo Research), and

RT-qPCRs were performed with the IP2 and E primers and probes as described above.

## Transcriptomics analysis in golden hamsters' brainstems

Two RNA-seq studies were conducted: (1) at 4 dpi (acute COVID-19), the infected group was composed of 5 male and 5 female hamsters infected with Wuhan and collected at 4 dpi, whereas the mock-infected group was composed of 3 male and 3 female mock-infected hamsters with samples processed at the same time; and (2) at 80 dpi (long Covid), the infected group was composed of 4 male and 4 female hamsters infected with Wuhan and collected at 80 dpi whereas the mock-infected group was composed of 4 male and 4 female mock-infected hamsters processed at the same time.

RNA preparation was used to build libraries using an Illumina Stranded mRNA library Preparation Kit (Illumina, USA) following the manufacturer's protocol. Quality control was performed on an Agilent BioAnalyzer. The Illumina NovaSeq X sequencer was used to produce paired-end 150b reads from the sequence libraries.

The RNA-seq analysis was performed with Sequana 0.15.3[90]. We used the RNA-seq pipeline 0.17.2 (https://github.com/sequana/sequana_rnaseq) built on top of Snakemake 7.32.4[91]. Briefly, reads were trimmed from adapters using Fastp 0.22.0[92], then mapped to the golden hamster MesAur1.0.100 genome assembly from Ensembl using STAR 2.7.10b[93]. FeatureCounts 2.0.1[94] was used to produce the count matrix, assigning reads to features using the corresponding annotation from Ensembl with strand-specificity information. Quality control statistics were summarized using MultiQC 1.11[95]. Statistical analysis on the count matrix was performed to identify differentially regulated genes. Clustering of transcriptomic profiles was assessed using a Principal Component Analysis (PCA). Differential expression testing was conducted using DESeq2 library 1.34.0[96] scripts, indicating the significance (two-sided test, adjusted p-values after Benjamini-Hochberg multiple testing correction, considering a false discovery rate FDR < 0.05) and the effect size (fold-change) for each comparison.

Finally, enrichment analysis was performed using modules from Sequana. The GO (Gene Ontology Consortium 2021) enrichment module uses PantherDB[97] and QuickGO[98] services. The KEGG pathways enrichment uses GSEApy 1.0.4[99], EnrichR[100], and KEGG database[101]. Both enrichments are one-sided tests, not considering potential depletions, and were corrected for multiple testing using Benjamini-Hochberg correction. Programmatic accesses to online web services were performed via BioServices 1.11.2[102].

## Quantitative PCR from golden hamsters' brainstems during long COVID

To validate the RNA-seq results, 25 genes from the main representative KEGG pathways were selected for quantifying host transcripts in brainstem samples from male and female hamsters infected with Wuhan or by the variants Delta and Omicron/BA.1 collected at 80 days after infection. Briefly, RNA was reverse transcribed to first-strand cDNA using the SuperScript™ IV VILO™ Master Mix (11766050, Invitrogen). qPCR was performed in a final volume of 10 µL per reaction in 384-well PCR plates using a thermocycler (QuantStudio 6 Flex, Applied Biosystems) and its related software (QuantStudio Real-time PCR System, Design and Analysis Software v. 2.7.0, Applied Biosystems). In each well, 2.5 µL of cDNA (12.5 ng) was added to 5 µL of Power SYBR Green PCR Master Mix (4367659, Applied Biosystems) and 2.5 µL of nuclease-free water containing 1 µM of golden hamster's primer pairs (Supplementary Table 1). The amplification conditions were as follows: 50 °C for 2 min, 95 °C for 10 min, and 45 cycles of 95 °C for 15 s and 60 °C for 1 min, followed by a melt curve from 60 °C to 95 °C. The *ACTB* and the *HPRT* genes were used as references. Variations in gene expression were calculated as the n-fold change in expression in the tissues from the infected hamsters compared with the tissues of the mock-infected group using the $2^{-\Delta\Delta Ct}$ method[103].

## Histopathology and immunohistochemistry

Samples destined to histopathology were collected and processed as follows: the brains were extracted from the skull and fixed in one piece. To obtain intact nasal turbinates, the heads of the animals were fixed in one piece after removal of the brain, the mandible and the skin. After fixation, the fragments were incubated with Osteomoll solution (101736, Merck) for 1 day followed by EDTA 0.5 M for 4 days to allow bone decalcification; afterwards, the nasal fragments were sectioned either in sagittal in transversal (Supplementary Fig 1). All samples were fixed in 10% neutral-buffered formalin for 24-48 hours and embedded in paraffin. Four-µm-thick sections were cut and stained with hematoxylin and eosin staining. Sections were cut to 4 µm thickness and processed for routine histology using hematoxylin and eosin staining and choromogenic ImmunoHistoChemistry (IHC) with hematoxylin counterstaining. IHC analysis was performed using the primary antibodies goat anti-GFAP (PA1-10004, Invitrogen; dilution 1:500) to detect astrocytes and rabbit anti-Iba-1 (019-19741, Wako; dilution 1:1000) to detect microglia. All slides were scanned using the AxioScan Z1 (Zeiss) system, and images were visualized with the Zen 2.6 software. We quantified the microglia and astrocyte staining using the QuPath software (version 0.5.1)[104]. Briefly, we defined regions of interest (ROIs) by drawing a 0.2 mm² rectangle in five brain areas of each sample (olfactory bulb, thalamus, midbrain, pons, and medulla oblongata). The area of Iba-1 or GFAP positive staining was calculated within the ROIs, and the results were expressed as percentage of positive stained area.

## Behavior tests

The tests were realized in isolators in a Biosafety level-3 facility that was specially equipped. We performed 3 sessions of behavioral tests starting at 14, 28 and 76 dpi. At each session, the tests were performed always in the same order, one test/day: light/dark box test (14, 28 and 76 dpi), sucrose splash test (15, 29 and 77 dpi), novelty-suppressed feeding test (16, 30 and 78 dpi), and novel object recognition (17, 31 and 79 dpi). We performed four batches of behavioral test: three repetitions with Wuhan, and one repetition with Delta and Omicron/BA.1. In each repetition of the experiment, both infected and mock-infected animals were included, and the testing of these animals was always conducted simultaneously in separate BSL-3 isolators. The behavior tests were performed during the 14-hours of light.

**Light/dark box test.** This test was based on a published protocol[105] with a few modifications. The light/dark box apparatus consisted of a box with two compartments of the same size: a dark chamber (black walls with an upper lid) and a light chamber (white wall, no upper lid). The chambers, which dimensions were adapted to fit an BSL-3 isolator (32 ×25 x 18 cm) were connected by a door (5 × 5.5 cm) in the middle of the wall. A standard commercial camera (C920 HD Pro, Logitech) was positioned around 30 cm above the white box, connected to a computer. A standard commercial 8 W LED lamp was also positioned around 30 cm above the white box to increase lighting (-800 lux). The animals were individually placed in the white box and were allowed to freely explore the light/dark box for 6 min. The boxes were cleaned with 70% ethanol between trials to avoid olfactory stimuli. One main output: the total time spent in the dark chamber, and two secondary outputs: the number of whole-body transitions from one chamber to another and the latency to enter the dark chamber were annotated manually by analyzing the recorded videos. Increased time spent in the dark box corresponds to anxious behavior.

**Sucrose splash test.** The test was based on an available protocol[106]. Briefly, the hamsters were isolated in their home cage and sprayed on the dorsal coat with a 10% sucrose solution (3 squirts/animal). The total time grooming was considered the main output of the test, and the latency to start grooming and the number of grooming sessions were

considered as secondary outputs. These variables were recorded manually over 5 min using a chronometer. Time grooming corresponds to an index of motivational and self-care behavior and decreased grooming time is considered as a depression-like behavior.

**Novelty-suppressed feeding test.** This test was adapted from a published protocol[107]. The test was held in a transparent arena (37 × 29 x 18 cm) with clean standard bedding. A filter paper disc (diameter of 12 cm) was placed at the center of the arena, and a food pellet was placed in the middle of the paper disc. A standard commercial 8 W LED lamp was positioned around 30 cm above the arena to increase lighting (~800 lux). One day before testing the hamsters were fasted. On the day of the test, the hamsters were individually placed in one corner of the arena. The test lasted a maximum of 10 minutes, and the latency to eat the food (defined as the time to grasp the food pellet with forepaws and bite, and not only approaching or sniffing the food pellet) was recorded using a chronometer. As soon as the food was eaten, the hamsters were removed from the arena and placed back into their home cage with *ad libitum* access to food. If the animal did not eat the food after 10 minutes, the latency to eat was considered to be 10 minutes for statistical purposes. Increased latency to eat indicates hyponeophagia and corresponds to an anxious behavior.

**Novel object recognition test.** This test was adapted from a published protocol[108]. The test was held in a transparent arena (37 ×29 x 18 cm) with clean standard bedding and consisted of two trials, filmed by a standard commercial camera (C920 HD Pro, Logitech) positioned around 30 cm above the arena. In trial 1, the animals were allowed to explore the arena with two identical objects placed at an equal distance in the middle of the arena (configuration ●-● or ▲-▲). The objects were 3D-printed green cylinders (diameter 4 cm, height 4 cm) and pink triangular prisms (base 3.8 cm, height 3.3 cm, length 4 cm). To avoid object preference, half of the animals of each experimental group were exposed to two green cylinders and the other half to two pink prisms. Further, the objects were cleaned with 70% ethanol between trials to avoid olfactory stimuli. The animals were placed in a corner of the arena and filmed during 5 min. To assess short-term memory, after 45 min of rest in the home cage, in trial 2, the animals were allowed to explore the same arena, this time containing one familiar object (the same as in trial 1) and one novel object (configuration ●-▲). Consequently, in trial 2, the arenas contained one green cylinder, and one pink prism placed at an equal distance in the middle of the arena. The animals were placed in a corner of the arena and filmed during 5 min. The exploration time (defined as the time sniffing or touching the object, but not sitting on it) was measured manually by analyzing the recorded videos. The time exploring the known object (A) and the novel object (B) was computed and the discrimination index was calculated according to the formula {B−A / B + A}. Decreased discrimination index corresponds to less time spent exploring the new object and is indicative of short-term memory impairment.

**Statistical analysis**

Behavioral tests described above were analyzed using mixed-effect model in which a random effect term accounted for the within-individual correlation arising from the longitudinal study design. The fixed effects corresponded to the set of available covariates, i.e., hamster sex, SARS-CoV-2 strain and days post-inoculation, along with the corresponding interactions should they significantly modulate the response. These analyses were performed using R v.4.4.1[109] with packages lme4 (v1.1-35.3)[110] and emmeans (v1.10.1)[111]. Other statistical analyses were performed using Prism 10 (GraphPad, version 10.2.3, San Diego).

**Reporting summary**

Further information on research design is available in the Nature Portfolio Reporting Summary linked to this article.

## Data availability

Source data are provided with this paper. The RNA-seq data generated in this study have been deposited in the Array Express database under accession code 4dpi: ArrayExpress E-MTAB-14779; 80 dpi: ArrayExpress E-MTAB-14780 [https://www.ebi.ac.uk/biostudies/arrayexpress]. Source data are provided with this paper.

## Code availability

The mixed-effect model analyses for behavioral tests generated in this study have been deposited in the Git repository [https://gitlab.pasteur.fr/hub/19421-long-covid-brainstem].

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

## Acknowledgements

The SARS-CoV-2 strain was supplied by the National Reference Centre for Respiratory Viruses hosted by Institut Pasteur (Paris, France) and headed by Pr. Sylvie van der Werf. The human sample from which strain 2019-nCoV/IDF0372/2020 was isolated has been provided by Dr. Xavier Lescure and Pr. Yazdan Yazdanpanah from the Bichat Hospital (Paris, France). The isolate SARS-CoV-2 Delta/2021/I7.2 200 (Delta, GISAID ID: EPI_ISL_2029113) and the isolate SARS-CoV-2 Omicron/B.1.1.529 (Omicron/BA.1, GISAID ID: EPI_ISL_6794907) was supplied by the Virus and Immunity Unit hosted by Institut Pasteur and headed by Dr. Olivier Schwartz. This work was supported by the Fondation pour la Recherche Médicale (grant ANRS MIE 202112015304) and by the Institut Pasteur's Programme Fédérateur de Recherche 4 (PFR-4 – Long COVID). This study has been labeled as a National Research Priority by the French National Orientation Committee for Therapeutic Trials and other research on COVID-19 (CAPNET). A.C. acknowledges funding from the Institut Pasteur's 2022-2023 Brain Axis SRA3 M2 Master Student Call. We thank Grégory Inizan for his help in producing customized apparatus for hamster behavioral testing, as well as Gautier Penchinat for 3D-printing the objects used in the novel object recognition test. We would like to thank Marion Berard, Laetitia Breton, Rachid Chennouf, Hamidou Diakhate, Eddie Maranghi and Mathilde Dubot for their help in implementing animal behavior tests in the Institut Pasteur animal facilities. We acknowledge Yakov Vitrenko, Biomics Platform, C2RT, Institut Pasteur, supported by France Génomique (ANR-10-INBS-09) and IBISA. We also thank Esma Karkeni, Single Cell Biomarkers Unit of Technology and Service (scBiomarkers UTechS), Institut Pasteur, for support with the SIMOA assays. Finally, we would like to thank Pr. Sandie Munier for proofreading the manuscript.

## Author contributions

Conceptualization: A.C., G.D.M and H.B. Methodology: G.D.M., F.L., T.O. and E.K. Investigation: A.C., F.L., L.K., M.T., G.D.M. Funding Acquisition: G.D.M., and H.B. Resources: G.D.M., F.L., D.H. and E.K. Supervision: G.D.M. Writing – Original Draft: A.C. and G.D.M. Writing – Review & Editing: all authors.

## Competing interests

The authors declare no competing interests.
