## [Transparent Peer Review file · Nature Communications]

Hamsters with long COVID present distinct transcriptomic profiles associated with neurodegenerative processes in brainstem

Corresponding Author: Dr Guilherme de Melo

Version 0:

Reviewer comments:

Reviewer #1

(Remarks to the Author)

Coleon et al. present a manuscript on long term consequences of SARS-CoV-2 in the well-established hamster model. The authors present strain and sex-specific effects on behavior and pathology up to 80 days post infection, which could be considered a long COVID period. This is important as long COVID is still often regarded as condition not based on virus-induced alterations of the brain, despite increasing evidence. Major and minor comments:

1. Please provide more data and evidence on the successful infectious virus isolation up to 80 dpi, currently only shown as red squares in Fig. 1. How about viral protein? How about infectious virus from other brain regions and organs? It would be important to clarify that infectious virus is persistent only in the brainstem, which is quite remarkable, especially as it is not expected to replicate, correct? Did you check in fat, which was described as a reservoir?
2. Iba1 and GFAP need to be quantified across dpi, sex and brain region, currently the data is descriptive.
3. There was no tissue damage and Nfl was unchanged at 80dpi, thus the authors do not provide direct evidence for neurodegeneration. N=4 is very low, especially given that these are hamsters with high inter-individual variability.
4. Gene expression alterations show dysregulation, which apparently does lead to dysfunction (behavior) in the absence of overt neurodegeneration (however, Nfl in n=4 hamsters..). However, as there is no protein data such as western blot or immunohistochemistry for neurodegeneration markers or neuronal integrity, conclusions are limited. Such limitations should be addressed and clarified in the abstract and conclusions.
5. Behavioral assessments are important and interesting, however, the authors did not perform pre-infection tests, limiting conclusion. For instance in the novel object there is considerable variability and no progression of deficits, without a pre-test it remains unclear whether differences are caused by the virus. Wuhan provides the most robust data, Delta and omicron effects are based on n=4/group, which for behavior in hamsters (not inbred mice) is severely underpowered.
6. How do these tests relate to pathology in the brain stem? Would the results not suggest additional yet undetected pathology outside the brain stem?
7. Could olfactory and gustatory deficits have impacted the behavior (relevant to long COVID)?
8. Do the authors have an explanation for wuhan infected hamsters preferring the familiar object (Zero would be no preference). Did you repeat the test with the same objects, which in that case lost novelty? The behavioral results need to be described and discussed in more detail.
9. There appear to be some language issues, some of which may lead to misunderstandings, e.g. title: Hamsters...exhibits, Abstract: after 80 days of infection (should be 80 days after infection?), Results: important body weight loss (significant/relevant loss?)..

Reviewer #2

(Remarks to the Author)

General Overview

The research investigates the long-term impact of SARS-CoV-2 infection in the brain using the Syrian golden hamster model. The research is relevant given the high prevalence of Long COVID and its ongoing impact on patients. A key strength of the study is the histological, immunohistochemical, and transcriptomic analyses, particularly focusing on the

brainstem at 4 and 80 dpi with three different viral isolates. However, there are several weaknesses that, if addressed, could significantly improve the study. Moreover, while behavioral tests were conducted, the absence of mechanistic experiments limits the ability to establish causal relationships, weakening the overall conclusions. Additionally, a thorough revision of the text is necessary, as sometimes it is very confusing and there are multiple overstatements where the data do not fully support the claims. These should be carefully rephrased to reflect the findings accurately. Moreover, several relevant references are missing, and the discussion lacks sufficient depth and comparison with previous studies, particularly those analyzing brain samples from both patients and hamsters. Given these limitations, the weaknesses currently outweigh the strengths, significantly diminishing the impact and enthusiasm for the study in its present form.

Other comments

- Title: If we consider the Long COVID definition by NASEM (as mentioned in line 343), "an infection-associated chronic condition that occurs after SARS-CoV-2 infection and is present for at least 3 months as a continuous, relapsing and remitting, or progressive disease state that affects one or more organ systems", can we conclusively state that hamsters develop Long COVID? A more appropriate phrasing would be "A Possible Model of Long COVID."
- Abstract: The abstract is too short and does not fully summarize the study, discuss relevant literature, or emphasize the importance of the work.
- Abstract, Line 32: "Deciphered" or "fully understood" would be more appropriate than "decrypted."
- Replicative Virus: Was replicative virus truly found? Further clarification is needed (See below).
- Abstract, Line 36: "Infected hamsters presented a neurodegenerative signature in the brainstem, with overexpression of innate immunity genes, impacted dopaminergic and glutamatergic synapses, and altered energy metabolism." The phrasing is unclear and potentially misleading; it should be revised for clarity.

Highlights

- "Transcriptomic profiles" or "transcriptome profiling" would be more accurate than "transcriptome profiles."

Introduction

- The introduction is confusing and lacks a clear focus on the research objectives and significance. Many sentences are too long and should be refined.
- Line 61: What is meant by "normal health"? Should this be "complete recovery", "full resolution of symptoms", or another term?
- Lines 63–68: The definition of Long COVID is stated twice, which is redundant and should be streamlined.
- Line 69: Stating that SARS-CoV-2 is definitively neuroinvasive may be an overstatement. A more cautious approach would be: "SARS-CoV-2 has the potential to infect the brain in some patients...."
- Line 71: The phrase "as the virus and inflammation are frequently observed in the olfactory mucosa"—does this definitively confirm neuroinvasion? Further clarification is needed.
- Line 74: "Changes in myelination and synapse organization"—what specific changes in myelination? Be more precise.
- Line 77: "Signaling defects in specific neuronal populations"—which specific defects and populations? More precision is needed.
- Line 78: What is meant by "first Long COVID consultation"? This should be described more clearly.
- Lines 79–81: "Viral persistence and a pro-inflammatory status may lead to altered energy and neurotransmitter metabolism (Refs 16, 17), all possible factors contributing to neuropsychiatric and cognitive symptoms (Refs 18–21)."
- This statement is confusing and should be rephrased for clarity. Additionally, Ref 16 is not related to SARS-CoV-2 but to DENV, and Ref 17 is, "as reviewed". These citations need to be reconsidered.
- Line 82: Why "neuroanatomic"? Would "molecular" or "cellular mechanisms" be more appropriate?
- Line 83: The words "infection" and "infected"* are overused throughout the text—consider varying the phrasing.
- Line 85: "An active infection and an intense inflammatory response." The findings do not fully support this statement. Regarding "active viral infection" (Line 139): The viral titers in the brainstem were below the quantification limit ($\sim 10^2$ TCID₅₀/mL), nevertheless, we could isolate infectious virus from 96.8% (31/32) of the brainstems from infected hamsters at 4 dpi." This raises several questions: What is the quantification limit? In Materials and Methods (Line 583) "brainstem samples were processed and added to VERO-E6 cells for 3 days. However, in Line 591, it is stated that "Viral titers were below the quantification limit ($\sim 10^2$ TCID₅₀/mL), although some wells in the titration plates presented cytopathic effects." Samples were "then passaged into VERO-TMPRSS2 cells for another 3–7 days followed by PCR". Why were VERO-

TMPRSS2 cells not used initially? The methodology suggests that fully mature, replicating viral particles were not observed using PFU or TCID₅₀ assays. A clearer explanation is needed, as the data do not convincingly demonstrate the isolation of infectious, replicating virus from the brainstem.

- Also, in Line 140, "we could isolate infectious virus from 96.8% (31/32) of the brainstems from infected hamsters at 4 dpi." What does 31/32 refer to? The total number of animals used? The total number of animals used is not stated in Materials & Methods, nor is the number of experiment repetitions.

-Line 85 - Intense inflammatory response: The findings do not strongly support this claim. Apart from Day 4 histopathology and some gliosis, no evidence of intense inflammation is shown. There is no quantification of inflammation in images or using alternative methods.

- Line 85: "By cellular exhaustion in the late phase, characterized mostly by impacted neurotransmission and altered energy metabolism. What does "impacted" mean? Be more precise. Does cell exhaustion necessarily follow from these findings? Both aspects were evaluated using RNA-seq, but no metabolic functional assays were performed. More caution is needed in interpretations and overstatements.

- Line 92: "That Long COVID is a factual biological issue that follows acute infection." Be careful with strong statements. While Long COVID is well acknowledged, this study contributes to understanding its mechanisms but does not establish causality.

- Line 153: "Particular phenotypic glial pattern." What phenotypic pattern? GFAP and Iba1 immunohistochemistry may not be sufficient to support this claim. Quantification (either image-based or through flow cytometry) would strengthen the analysis.

-Line 181: "No difference was observed, except for one male hamster showing increased NFL levels in the serum." Is this an outlier? If so, it should not be mentioned. One out of how many animals? Not stated.

- Line 194: "From which 391 and 115 DEGs (increased or decreased, respectively) had a fold change higher than 2." P-values, adjusted values (Padj), or FDR must be included for proper statistical validation. A volcano plot and a Venn diagram of co-expressed genes by sex and dpi would improve the presentation.

- Line 272: "Detected ten of them differentially expressed in the brainstem at 4 dpi (ATXN1, HTT, KIF5A, LRRK2/PARK8, MAPT, PARK7/DJ-1, PSEN2, SQSTM1, SOD1, TBK1)." The phrasing is unclear and should be revised for clarity. A comparison with human public datasets would significantly strengthen this analysis.

- Lines 299–300: "To check if altered brainstem metabolism due to SARS-CoV-2 infection would be enough to cause clinical manifestations related to Long COVID." Assuming a direct correlation between behavioral changes and brainstem metabolism is an overstatement. This should be rephrased to reflect the limitations.

- Statistics of behavioral tests should be depicted in all graphs.

Reviewer #3

(Remarks to the Author)

I. Key Results

Coleon et al. employ the golden hamster model to investigate both the acute and chronic phases of SARS-CoV-2-induced CNS symptoms. The study is comprehensive, presenting clinical data, histology, RNAseq, and in vivo behavioral data following infection with three SARS-CoV-2 strains (Wuhan, Delta, and Omicron). Notably, the model reveals sex-dependent differences both during acute infection and in the post-infection phase. A particularly striking and novel finding is the persistence of low numbers of virus in the brain, especially the brainstem, up to 80 dpi, which could potentially underlie long-term neurological consequences. RNAseq data suggests a broad range of changes in the brain upon infection with the Wuhan strain, which could help decipher the processes involved in long covid.

II. Validity

Overall the manuscript appears to be comprehensive and the data presented valid. However, the behavioral data set should be readdressed and reinterpreted.

III. Figure-by-Figure Remarks

Below are remarks of things that should be addressed:

Figure 1 and related: Clinics and virus

Line 104-106: The claim that "female hamsters also exhibited important body weight loss but milder clinical scores compared to males (Fig.1E, F)" should be supported by statistical comparisons (e.g. direct comparisons of clinical scores between sexes).

Figure legends should show what is presented in the panels in terms of mean/SEM for B and E and the median or mean (?) for C and F. Showing some kind of range would be nice for the score data.

Lines 112-114: The statement regarding weight gain from 10 dpi to 80 dpi, where infected males did not reach the weight of the control group, requires appropriate statistical support.

Lines 116-120: The authors note a higher LW/BW ratio in some groups, suggestive of inflammation, edema, and congestion. It should be made clear that this was only a trend for the Wuhan group as shown in the figure.

Figure 2 and related: Histology

The authors show and describe myeloid and astrocytic changes in different brain regions. Astrocytic changes observed at 80 dpi (and 4 dpi) are highly interesting as there are as yet no data sets showing such late time points (Lines 172-177). I would like to see some kind of quantification to warrant that this is a true effect and if perhaps there is a sex effect as well. The acute Iba1 effect has already been described in the literature and is fine this way describing it.

Figure 3-5 and related: RNAseq

This is a highly valuable dataset for learning about and deciphering what happens post SARS-CoV-2 in the hamster brain and the presented changes are highly interesting.

The way the RNAseq data is presented is great this way, but the field would benefit from an additional look at the sex differences that are already spread all over the paper, but not at all for this data set. The $n = 4$ per sex should allow at least to a certain degree to see if males and females respond differently to the infection in the acute and long-term phase.

Figure 6 and related: Behavior

Behavioral Data – Novel Object Recognition (NOR)

The raw NOR data do not seem to fully support the claim of persistent memory impairment in both Wuhan-infected males and females (lines 332-333). Specifically:

Supplementary Figure 11 shows that Wuhan-female hamsters have a significantly decreased discrimination index at day 79, while other female groups are similar to mock controls.

In male hamsters, there is only a trend for decreased DI at 79 dpi, though an early effect is noted.

Panels D and G of Figure 6 appear identical between males and females, which raises concerns regarding possible misplacement of panels given that supplemental data suggest sex differences.

Behavioral Data – Hyponeophagia and Sucrose Splash Test

Hyponeophagia:

In the raw data (Fig Sup 8) we see a clear effect in males, but not in females. While males at 16 dpi show reduced anxiety for Delta (and a trend for such a decrease for Omicron) Wuhan males are not different from mock animals. There is a clear increase at 78 dpi for the Wuhan animals.

Looking at Panels B and E of Figure 6 suggests that females show an anxiety phenotype at 78 dpi which is not seen in the raw data and hence may not be reliable, while the male 78 dpi effect for Wuhan is credible.

Sucrose Splash Test:

The text (lines 325-328) should be rephrased for clarity, after mentioning a dichotomy I would expect to read about the differences between males and females?

The term “depressive-like behavior” should be maintained rather than implying clinical depression in hamsters.

IV. Significance

3. Significance

Novelty and Impact:

The persistent presence of viral RNA in the brain up to 80 dpi is a novel observation that contrasts with other studies reporting viral clearance within days or weeks (e.g. Frere et al.). This finding may offer insight into potential mechanisms underlying long-term neurological and behavioral changes.

The extensive dataset—ranging from molecular to behavioral measures—provides valuable information for the field. However, given that several aspects of the data (e.g., the behavioral readouts) are interpreted with advanced statistical models that do not always align with raw data trends, further validation is needed before drawing broad conclusions.

The behavioral outcomes are interesting and after getting them into a better shape will hopefully reveal an interesting long-term phenotype of “post covid hamsters.”

V. Data and Methodology

Figure Presentation and Statistical Details

Figure Legends:

Clarify the statistical presentation in Figures 1 and 6. For instance, in Figure 1, panels B and E should indicate whether values are represented as mean \pm SEM, and panels C and F should specify whether median values or means are used, ideally including a range for clinical score data.

Cohort Consistency and Controls

It is unclear if PBS/mock controls were tested simultaneously with each strain or pooled from different experimental batches. The manuscript should state whether experiments with Wuhan, Delta, and Omicron/BA.1 were conducted concurrently. If not, the methods (and potentially the supplementary figures) should account for this (e.g., via a 2-way ANOVA testing for both cohort and virus effects).

Behavioral Testing Conditions

The methods should specify the lighting scheme (e.g., light intensities in lux in the center vs. border of the NOR arena), the time of day when tests were conducted, and report the velocity/distance moved by hamsters during tests. This information is critical, as hypo- or hyperactivity can influence behavioral outcomes.

Additional video-based analyses (e.g., time spent in center vs. border zones) could help distinguish between memory impairment and anxiety-related behaviors, especially for the NOR video dataset.

Code and Statistical Transparency

Given the use of advanced statistical analyses (e.g., mixed model regression), the authors should provide the analysis code in a supplementary file. Additionally, a supplementary table summarizing effect sizes and/or p values would aid in evaluating the robustness of the findings.

VI. Analytical Approach

Strength of Statistical Methods:

The RNAseq data presentation is well done, but the analysis would benefit from an exploration of sex-specific responses, especially given the $n = 4$ per sex in both the acute and long-term phases.

The advanced statistical methods (mixed model regression) used to account for inter-animal variance in the behavioral tests are appropriate; however, the discrepancies between these modeled data and the raw data (particularly in the NOR test) necessitate a re-examination of the analysis approach to ensure that findings are not overstated.

VII. Suggested Improvements

Statistical Reporting:

Include direct statistical comparisons between male and female groups where claims are made (e.g., body weight changes, clinical scores).

Clarify and standardize the presentation of data in figures, explicitly stating whether the data are means, medians, ranges, etc.

Provide a supplementary table of effect sizes/p values for the behavioral analyses.

Data Consistency and Validation:

Revisit the NOR data analysis (and all the data to be sure) to resolve the inconsistencies between raw data (including supplementary figures) and the main figures.

Double-check Figure 6, Panels D and G to ensure the correct data is presented for each sex.

Include quantification of astrocytic changes (at 4 dpi and 80 dpi) to confirm the reported effects and assess potential sex differences.

Additional Experimental Suggestions:

It would strengthen the manuscript to include histological validation (e.g., immunostaining) of the mRNA changes at 80 dpi for markers relevant to neurodegeneration or inflammation (e.g., TH, alpha-Synuclein, dopamine or glutamate receptors, IFN-related proteins).

An additional deeper look at the RNAseq data in terms of sex effects/differences at 4 and 80 dpi.

Velocity or distance moved of the hamsters during the behavioral tests should be analyzed and shown as the activity of the animals largely influences the outcomes of the tests (e.g., performers and non-performers, do animals of certain groups explore less or more due to hypo- or hyperactivity?).

Light intensities for the behavioral tests and zones need to be reported (xxx lux in center of the NOR arena, xxx lux in the border zone etc.).

Methodological Details:

Specify whether all behavioral tests were performed in the same experimental batch and under identical conditions, e.g., were the different virus groups different batches?

Were the PBS controls in the same experimental tests at the same time? Were they pooled in the analyses from different batches?

If the Wuhan, Delta, and Omicron experiments were not performed at the same time, this should be stated in the methods section. The supplementary figures should show this somehow, as the mock controls are currently pooled (e.g., a 2-way ANOVA could be used for testing cohort and SARS-CoV-2 effects if the data allows this in terms of distribution).

Detail the lighting conditions, light scheme, and test timing, as these factors can significantly impact hamster behavior.

Terminology:

Maintain the use of "depressive-like behavior" rather than suggesting clinical depression in hamsters.

Clarify the terminology referring to "parkinsonism" versus "Parkinson's disease."

Code Availability:

Provide the code used for statistical analyses of the behavioral data (regression) in a supplementary file to facilitate transparency and reproducibility.

7. Clarity and Context

Manuscript Accessibility

The manuscript is generally accessible but would benefit from additional context in several areas:

Clear differentiation between trends and statistically significant effects.

Detailed figure legends that allow the reader to interpret mean/median values, SEM, ranges, and effect sizes.

A more thorough description of the behavioral testing environment (e.g., lighting, time of day, arena conditions) to help contextualize the findings.

Context within the Literature

Provide a discussion that situates the persistent brain virus findings within the broader literature, including contrasting studies where viral persistence was not observed.

Consider discussing the role of SNCA in SARS-CoV-2 infection and neurodegeneration, either by referencing existing literature or by indicating if the RNAseq data provide insights.

Some of the changes observed in the RNAseq dataset are in line, others contradicting to the 31 dpi dataset of Frere et al. It would be good to see more discussion on this.

X. Conclusion

The study presents valuable and novel findings using a well-established hamster model, particularly regarding long-term viral persistence and sex-dependent differences. However, several aspects of the data presentation and statistical analysis need to be clarified and strengthened before the manuscript can be considered for publication. I recommend that the authors address the points listed above, providing additional statistical details, clarifying methodological conditions, and ensuring consistency between raw and analyzed data.

The discrepancies between the raw data and the regression-modeled data raise concerns about the overstatement or validity of some findings (e.g., for the NOR). While the NOR data warrants skepticism and may require alternative presentation, calculation, and double-checking, the behavioral experiments conducted in a BSL3 facility with hamsters represent a significant achievement given the inherent complexities. Moreover, the RNAseq dataset from the 80 dpi timepoint of the brainstem is of high interest to the community.

The presentation of data without explicit p values, relying on reader estimation through CI overlaps, poses a challenge. A supplementary table with effect sizes or p values could enhance trust in the model.

Version 2:

Reviewer comments:

Reviewer #1

(Remarks to the Author)

The authors have addressed some of my points, however, I strongly suggest to add a limitations section to the discussion, summarizing the limitations discussed with the reviewers. The lack of behavioral effects may be explained by $n=4$, which is way too low for these kinds of behavioral tests with hamsters, and not by the variant. N for all groups should always be stated in the figure legend.

Reviewer #2

(Remarks to the Author)

The manuscript has been substantially improved in accordance with this reviewers' suggestions. The major concerns previously raised—particularly those regarding viral detection and isolation from the brain, as well as the quantification of astrocytic (GFAP) and microglial (Iba-1) activation, have been adequately addressed. In addition, the text has been thoroughly revised, resulting in a clearer and more cohesive presentation. From this reviewer's perspective, the manuscript is now suitable for publication.

Reviewer #3

(Remarks to the Author)

Dear Authors,

Thank you for your diligent work on the revised manuscript. Your comprehensive responses to the reviewers' comments have significantly enhanced the manuscript's clarity, robustness, and impact. We particularly commend your efforts in providing new immunofluorescence data, quantifying glial changes, refining terminology, and making statistical analyses publicly available.

There is one minor point regarding Figure 6 (Long-term impact of SARS-CoV-2 infection on neuropsychiatric and cognitive behavior) that we believe could benefit from a bit more clarification for the readership.

• Figure 6 Model Clarification: Panels B, C, D, E, F, and G of Figure 6 display average predicted values derived from mixed-model regressions. To ensure maximum transparency and prevent any potential misinterpretation or questions from readers, it is essential to explicitly state the specific mixed-effect model used for each behavioral test within the Figure 6 legend, or directly on the relevant panels themselves.

- For the Novelty-suppressed feeding test (Panels B and E), the model used was Day * Sex * Infection.
- For the Sucrose splash test (Panels C and F), the model used was Day * Sex * Infection.
- For the Novel object recognition test (Panels D and G), the model used was Day + Sex + Infection.

- Clarifying these precise model specifications will provide readers with a clearer understanding of the data presentation, especially noting how factors like sex and time are incorporated into the predictions. We also recommend explicitly referring to the Git repository (<https://gitlab.pasteur.fr/hub/19421-long-covid-brainstem>) as the source for the complete statistical analyses and model specifications in the legend. This will ensure full transparency and further underscore the rigor of your statistical approach.

Once this small but important clarification for Figure 6 is incorporated, the manuscript is well-prepared for final submission.

Minor detail: p values in the manuscript should be written with a “.”.

Point-by-point answers to the Reviewers

We appreciate the careful review of our manuscript by the reviewers. Their constructive comments and suggestions were much appreciated, and the new version of the manuscript has benefited considerably by this help. In the following sections, we address the comments made. Please note that all changes made in the text are highlighted in red.

Reviewer #1 (Remarks to the Author):

Coleon et al. present a manuscript on long term consequences of SARS-CoV-2 in the well-established hamster model. The authors present strain and sex-specific effects on behavior and pathology up to 80 days post infection, which could be considered a long COVID period. This is important as long COVID is still often regarded as condition not based on virus-induced alterations of the brain, despite increasing evidence. Major and minor comments:

R: We would like to thank the reviewer for the comments and suggestions made in our manuscript.

1. Please provide more data and evidence on the successful infectious virus isolation up to 80 dpi, currently only shown as red squares in Fig. 1. How about viral protein? How about infectious virus from other brain regions and organs? It would be important to clarify that infectious virus is persistent only in the brainstem, which is quite remarkable, especially as it is not expected to replicate, correct? Did you check in fat, which was described as a reservoir?

R: Initially, we proceeded to a classical TCID50 titration using Vero-E6 cells aiming at the quantification of the viral titer in brainstem homogenates of infected hamsters; this proved challenging: on the one hand, we detected lysed wells, on the other hand we did not obtain a consistent pattern of lysed wells, even in the first dilution, probably due to low viral load and/or high density of homogenized tissue. We could not quantify the titer, and we stated that it was below the quantification limit (line 641). To overcome this issue, we recovered the supernatant from the lysed wells from these TCID50s to try to amplify the virus on Vero-TMPRSS2 cells (data represented by the red squares in Figure 1). Now, to complete these data, we performed immunofluorescence to identify two viral proteins (spike and nucleoprotein) in Vero-E6 cells exposed to brainstem homogenates at 4 dpi and 80 dpi. These new data were included in Figure 1 and in Supplementary Figure 3 to support the message that the SARS-CoV-2 in the brainstem is still infectious, even if the viral load is low. In this study we focused on the brainstem, and we cannot exclude that the virus may also persist in other areas.

2. Iba1 and GFAP need to be quantified across dpi, sex and brain region, currently the data is descriptive.

R: We quantified Iba1 and GFAP according to dpi, sex and brain region. To facilitate reading, we kept GFAP data in Figure 2 and moved Iba-1 data in Supplementary Fig 5.

3. There was no tissue damage and Nfl was unchanged at 80dpi, thus the authors do not provide direct evidence for neurodegeneration. N=4 is very low, especially given that these are hamsters with high inter-individual variability.

R: We agree with the reviewer. As supported by histology (Supplementary Fig. 5) and by Nfl levels in the serum (Supplementary Fig. 5), we indeed did not find evidence of extensive tissue damage in the brain.

4. Gene expression alterations show dysregulation, which apparently does lead to dysfunction (behavior) in the absence of overt neurodegeneration (however, Nfl in n=4 hamsters..). However, as there is no protein data such as western blot or immunohistochemistry for neurodegeneration markers or neuronal integrity, conclusions are limited. Such limitations should be addressed and clarified in the abstract and conclusions.

R: We have not claimed that the hamsters suffer from neurodegeneration. Instead, we provide evidence that the animals exhibit a pattern of gene expression that can be associated with neurodegenerative processes. We call the "neurodegenerative signature" the group of the

following pathways: innate immunity, neurotransmission, energy metabolism, proteasome; that we found dysregulated in the brainstem of the infected hamsters, among the eight pathways proposed by Wilson et al. (2023). We rewrote the abstract (lines 27-44) and the conclusions (509-520) accordingly.

Wilson, DM, III et al. Hallmarks of neurodegenerative diseases. Cell 186, 693-714 (2023).

5. Behavioral assessments are important and interesting, however, the authors did not perform pre-infection tests, limiting conclusion. For instance in the novel object there is considerable variability and no progression of deficits, without a pre-test it remains unclear whether differences are caused by the virus. Wuhan provides the most robust data, Delta and omicron effects are based on n=4/group, which for behavior in hamsters (not inbred mice) is severely underpowered.

R: For the behavior tests, the infected and non-infected groups were performed simultaneously in two different isolators, at the same time. The only difference between the two groups was infection. The pre-infection tests are not relevant in our setting because we want to be able to strictly compare a group of infected animals with a group of control animals at the same age. Even if we show individualized data (Supplementary figures 10-13, Source data), the power and robustness of our analysis lies on the fact that we consider different factors at the same time: sex, viral variant, time post-infection. We added a link to the mixed-effect model analyses for behavioral tests in lines 872-873.

6. How do these tests relate to pathology in the brain stem? Would the results not suggest additional yet undetected pathology outside the brain stem?

R: Behavioral tests revealed anxiety, depression and alterations in short-term memory at 80dpi in infected hamsters. The brainstem among other regions of the CNS is known to play an essential role in these features. Further, transcriptional abnormalities linked to energy metabolism or synaptic transmission were found in this specific region. However, we agree with the reviewer. We cannot specifically relate these behavioral results with some brain regions. Therefore, to avoid misinterpretations, we rephrased the lines 328-334.

7. Could olfactory and gustatory deficits have impacted the behavior (relevant to long COVID)?

R: Olfactory and gustatory deficits can indeed affect behavior. However, in our previous experience, the anosmia induced by SARS-CoV-2 in hamsters is self-limiting and restricted to the acute phase (3-5 dpi) and no more signs of anosmia are observed at 14 dpi.

de Melo, GD et al. COVID-19-related anosmia is associated with viral persistence and inflammation in human olfactory epithelium and brain infection in hamsters. Science Translational Medicine 13, eabf8396 (2021).

8. Do the authors have an explanation for wuhan infected hamsters preferring the familiar object (Zero would be no preference). Did you repeat the test with the same objects, which in that case lost novelty? The behavioral results need to be described and discussed in more detail.

R: We did not repeat it with the same objects because this is outside the scope of the test. In this test, the time taken to explore the novel object provides an index of recognition memory. We did not compare individual performance or preferences toward the novel or the familiar object; the test compares how the infected groups behave relative to a mock infected group. The behavioral tests are described in the Methods section (lines 794-815).

9. There appear to be some language issues, some of which may lead to misunderstandings, e.g. title: Hamsters...exhibits, Abstract: after 80 days of infection (should be 80 days after infection?), Results: important body weight loss (significant/relevant loss?).

R: We thank the reviewer for this comment, and we tried to do our best to correct the language issues throughout the text.

Reviewer #2 (Remarks to the Author):

General Overview

The research investigates the long-term impact of SARS-CoV-2 infection in the brain using the Syrian golden hamster model. The research is relevant given the high prevalence of Long COVID and its ongoing impact on patients. A key strength of the study is the histological, immunohistochemical, and transcriptomic analyses, particularly focusing on the brainstem at 4 and 80 dpi with three different viral isolates. However, there are several weaknesses that, if addressed, could significantly improve the study. Moreover, while behavioral tests were conducted, the absence of mechanistic experiments limits the ability to establish causal relationships, weakening the overall conclusions.

R: We thank the reviewer for the careful review of our manuscript and for the comments and suggestions.

Additionally, a thorough revision of the text is necessary, as sometimes it is very confusing and there are multiple overstatements where the data do not fully support the claims. These should be carefully rephrased to reflect the findings accurately.

R: We performed a revision throughout the text to limit our claims to the findings.

Moreover, several relevant references are missing, and the discussion lacks sufficient depth and comparison with previous studies, particularly those analyzing brain samples from both patients and hamsters. Given these limitations, the weaknesses currently outweigh the strengths, significantly diminishing the impact and enthusiasm for the study in its present form.

R: We revised the text and added new references (references number 12-14, 16, 18-20, 26, 40, 42-48, 104). We consider that we have cited now the most adapted references in the field, including several studies that focus on human patients and on the hamster model.

Other comments

- Title: If we consider the Long COVID definition by NASEM (as mentioned in line 343), “an infection-associated chronic condition that occurs after SARS-CoV-2 infection and is present for at least 3 months as a continuous, relapsing and remitting, or progressive disease state that affects one or more organ systems”, can we conclusively state that hamsters develop Long COVID? A more appropriate phrasing would be "A Possible Model of Long COVID."

R: We are convinced (see also the first comment of reviewer 1 in his first paragraph) that we have all the necessary elements to attest that the hamsters used in this study have long Covid.

- They have persisting SARS-CoV-2 infection and altered immunometabolism in the brain.
- They have persisting and/or relapsing symptoms up to 80 days post infection
- This affects their behavior, and we provide data on how this induced pathology in different brain regions

For these reasons, we have decided to leave the title as originally proposed.

Regarding the definition, we decided to update the discussion with the most recent definition to date, proposed by the WHO on February 26, 2025: “*Long Covid is characterized by a range of symptoms that usually start within 3 months of the initial COVID-19 illness and last for at least 2 months. These symptoms may persist from the initial illness or develop after recovery. It can affect a person's ability to perform daily activities such as work or household chores and restrict social participation*” (lines 350-356). We consider that our results found in hamster are perfectly in line with this definition (with the limitations of an animal model).

WHO. *Post COVID-19 condition (long COVID)*. <https://covid19.who.int/> (2022).
[https://www.who.int/news-room/fact-sheets/detail/post-covid-19-condition-\(long-covid\)](https://www.who.int/news-room/fact-sheets/detail/post-covid-19-condition-(long-covid)) (2025).

- Abstract: The abstract is too short and does not fully summarize the study, discuss relevant literature, or emphasize the importance of the work.

R: We rewrote some parts of the abstract; however, Nature Communication imposes the abstract to be no longer than 200 words.

- Abstract, Line 32: "Deciphered" or "fully understood" would be more appropriate than "decrypted."

R: We thank the reviewer for his suggestion. We revised the abstract as requested (line 31-32).

- Replicative Virus: Was replicative virus truly found? Further clarification is needed (See below).

R: As we could amplify the virus, we can conclude that the virus is replicative. To further confirm this finding, we now provide immunofluorescence studies to identify two viral proteins (spike and nucleoprotein) in Vero-E6 cells exposed to brainstem homogenates at 4 dpi and 80 dpi. These new data were included in Figure 1 and in Supplementary Figure 3.

- Abstract, Line 36: "Infected hamsters presented a neurodegenerative signature in the brainstem, with overexpression of innate immunity genes, impacted dopaminergic and glutamatergic synapses, and altered energy metabolism." The phrasing is unclear and potentially misleading; it should be revised for clarity.

R: We revised the abstract as requested (lines 35-38).

Highlights

- "Transcriptomic profiles" or "transcriptome profiling" would be more accurate than "transcriptome profiles."

R: We replaced transcriptome profiles with transcriptomic profiles as suggested (line 53 and throughout the text).

Introduction

- The introduction is confusing and lacks a clear focus on the research objectives and significance. Many sentences are too long and should be refined.

R: We reorganized and rewrote the introduction as requested.

- Line 61: What is meant by "normal health"? Should this be "complete recovery", "full resolution of symptoms", or another term?

R: We rewrote this sentence for clarity (line 62).

- Lines 63–68: The definition of Long COVID is stated twice, which is redundant and should be streamlined.

R: We adapted this section as suggested (lines 65-66).

- Line 69: Stating that SARS-CoV-2 is definitively neuroinvasive may be an overstatement. A more cautious approach would be: "SARS-CoV-2 has the potential to infect the brain in some patients...."

R: We thank the reviewer for this very important comment that we fully agree with. Therefore, we rewrote this sentence as suggested (lines 68-69).

-Line 71: The phrase "as the virus and inflammation are frequently observed in the olfactory mucosa"—does this definitively confirm neuroinvasion? Further clarification is needed.

R: We rewrote this sentence for clarity (lines 70-72).

-Line 74: "Changes in myelination and synapse organization"—what specific changes in myelination? Be more precise.

R: We rewrote this sentence for clarity (lines 70-74).

-Line 77: "Signaling defects in specific neuronal populations"—which specific defects and populations? More precision is needed.

R: We rewrote this sentence for clarity (lines 70-77).

- Line 78: What is meant by "first Long COVID consultation"? This should be described more clearly.

R: It means first consultation at the long COVID clinic according to Salmon et al. (2024). We added more details in line 74-77.

*Salmon, D et al. Patients with Long COVID continue to experience significant symptoms at 12 months and factors associated with improvement: A prospective cohort study in France (PERSICOR). **International Journal of Infectious Diseases** 140, 9-16 (2024).*

- Lines 79–81: "Viral persistence and a pro-inflammatory status may lead to altered energy and neurotransmitter metabolism (Refs 16, 17), all possible factors contributing to neuropsychiatric and cognitive symptoms (Refs 18–21)."

- This statement is confusing and should be rephrased for clarity. Additionally, Ref 16 is not related to SARS-CoV-2 but to DENV, and Ref 17 is, "as reviewed". These citations need to be reconsidered.

R: We rewrote the paragraph and updated the references (line 78-86).

- Line 82: Why "neuroanatomic"? Would "molecular" or "cellular mechanisms" be more appropriate?

R: We focused our study on the brainstem, which hosts different nuclei related to cognition and mood, including for example, the substantia nigra, whose neurons produce dopamine. We agree that "molecular mechanisms" is also appropriate, and we add this in line 86.

-Line 83: The words "infection" and "infected"* are overused throughout the text—consider varying the phrasing.

R: We have changed the wording wherever possible.

- Line 85: "An active infection and an intense inflammatory response." The findings do not fully support this statement. Regarding "active viral infection" (Line 139): The viral titers in the brainstem were below the quantification limit ($\sim 10^2$ TCID₅₀/mL), nevertheless, we could isolate infectious virus from 96.8% (31/32) of the brainstems from infected hamsters at 4 dpi." This raises several questions: What is the quantification limit? In Materials and Methods (Line 583) "brainstem samples were processed and added to VERO-E6 cells for 3 days. However, in Line 591, it is stated that "Viral titers were below the quantification limit ($\sim 10^2$ TCID₅₀/mL), although some wells in the titration plates presented cytopathic effects." Samples were "then passaged into VERO-TMPRSS2 cells for another 3–7 days followed by PCR". Why were VERO-TMPRSS2 cells not used initially? The methodology suggests that fully mature, replicating viral particles were not observed using PFU or TCID₅₀ assays. A clearer explanation is needed, as the data do not convincingly demonstrate the isolation of infectious, replicating virus from the brainstem.

R: Initially, we proceeded to a classical TCID₅₀ titration using Vero-E6 cells aiming at the quantification of the viral titer in brainstem homogenates of infected hamsters; this proved challenging: on the one hand, we detected lysed wells, on the other hand we did not obtain a consistent pattern of lysed wells, even in the first dilution, probably due to low viral load and/or high density of homogenized tissue. We could not quantify the titer, and we stated that it was below the quantification limit (line 641). To overcome this issue, we recovered the supernatant

from the lysed wells from these TCID50s to try to amplify the virus on Vero-TMPRSS2 cells (data represented by the red squares in Figure 1). Now in the present version and to complete these data, we show immunofluorescence results identifying two viral proteins (spike and nucleoprotein) in Vero-E6 cells exposed to brainstem homogenates at 4 dpi and 80 dpi. These new data were included in Figure 1 and in Supplementary Figure 3 to support the message that the SARS-CoV-2 in the brainstem is still infectious, even if the viral load is low. In this study we focused on the brainstem, and we cannot exclude that the virus may also persist in other areas.

- Also, in Line 140, "we could isolate infectious virus from 96.8% (31/32) of the brainstems from infected hamsters at 4 dpi." What does 31/32 refer to? The total number of animals used? The total number of animals used is not stated in Materials & Methods, nor is the number of experiment repetitions.

R: We rephased these sentences for better clarity (lines 156-159 and 170-172). The number of animals is stated in each figure and/or figure legend, as well as in the excel source data file.

-Line 85 - Intense inflammatory response: The findings do not strongly support this claim. Apart from Day 4 histopathology and some gliosis, no evidence of intense inflammation is shown. There is no quantification of inflammation in images or using alternative methods.

R: We rephased this sentence. Further, we quantified IBA-1 and GFAP immunostainings according to dpi, sex and brain region. To facilitate reading, we kept GFAP data in Figure 2 and moved IBA-1 data in Supplementary Fig 5.

- Line 85: "By cellular exhaustion in the late phase, characterized mostly by impacted neurotransmission and altered energy metabolism. What does "impacted" mean? Be more precise. Does cell exhaustion necessarily follow from these findings? Both aspects were evaluated using RNA-seq, but no metabolic functional assays were performed. More caution is needed in interpretations and overstatements.

R: We agree with the reviewer that the term "impacted" is not correct. It has been now replaced by "dysregulated genes". We rephased this sentence to avoid overstatement (lines 90-91).

- Line 92: "That Long COVID is a factual biological issue that follows acute infection." Be careful with strong statements. While Long COVID is well acknowledged, this study contributes to understanding its mechanisms but does not establish causality.

R: We changed this sentence in lines 95-98 and 417-421.

- Line 153: "Particular phenotypic glial pattern." What phenotypic pattern? GFAP and Iba1 immunohistochemistry may not be sufficient to support this claim. Quantification (either image-based or through flow cytometry) would strengthen the analysis.

R: We rephased this subtitle (lines 179-180). We quantified IBA-1 and GFAP immunostainings according to days post-infection, sex and brain region. To facilitate reading, we kept GFAP data in Figure 2 and moved IBA-1 data to Supplementary Fig 5.

-Line 181: "No difference was observed, except for one male hamster showing increased NfL levels in the serum." Is this an outlier? If so, it should not be mentioned. One out of how many animals? Not stated.

R: We rephased this sentence (lines 187-190). We tested n=4 males and n=4 females. The number of animals is now represented in the graphs and indicated in the figure legend (Supplementary Figure 2).

- Line 194: "From which 391 and 115 DEGs (increased or decreased, respectively) had a fold change higher than 2." P-values, adjusted values (P_{adj}), or FDR must be included for proper statistical validation. A volcano plot and a Venn diagram of co-expressed genes by sex and dpi would improve the presentation.

R: The adjusted P-values are shown in Figures 3 and 4A (color scale). In Figures 4B-D and 5, the symbols *, **, *** denote $p < 0.05$, $p < 0.01$ and $p < 0.001$, respectively, according to the Benjamini–Hochberg-adjusted test. These analyses are described in the Methods section (lines 680-692). We also added an Upset plot highlighting differences in differentially regulated gene lists grouped by sex, days post-infection and regulation orientation of the genes (Supplementary Fig. 6).

- Line 272: "Detected ten of them differentially expressed in the brainstem at 4 dpi (ATXN1, HTT, KIF5A, LRRK2/PARK8, MAPT, PARK7/DJ-1, PSEN2, SQSTM1, SOD1, TBK1)." The phrasing is unclear and should be revised for clarity. A comparison with human public datasets would significantly strengthen this analysis.

R: R: We rewrote this sentence for clarity (lines 300-303).

- Lines 299–300: "To check if altered brainstem metabolism due to SARS-CoV-2 infection would be enough to cause clinical manifestations related to Long COVID." Assuming a direct correlation between behavioral changes and brainstem metabolism is an overstatement. This should be rephrased to reflect the limitations.

R: We agree with the reviewer and have rephrased this sentence (lines 328-334).

- Statistics of behavioral tests should be depicted in all graphs.

R: The behavioral analysis contains such a high number of p values that it is impossible to construct a comprehensible figure. To address this, the complete results are accessible in the Git repository at <https://gitlab.pasteur.fr/hub/19421-long-covid-brainstem> (lines 872-873).

Reviewer #3 (Remarks to the Author):

I. Key Results

Coleon et al. employ the golden hamster model to investigate both the acute and chronic phases of SARS-CoV-2-induced CNS symptoms. The study is comprehensive, presenting clinical data, histology, RNAseq, and in vivo behavioral data following infection with three SARS-CoV-2 strains (Wuhan, Delta, and Omicron). Notably, the model reveals sex-dependent differences both during acute infection and in the post-infection phase. A particularly striking and novel finding is the persistence of low numbers of virus in the brain, especially the brainstem, up to 80 dpi, which could potentially underlie long-term neurological consequences. RNAseq data suggests a broad range of changes in the brain upon infection with the Wuhan strain, which could help decipher the processes involved in long covid.

R: We thank the reviewer for this comprehensive analysis of our study.

II. Validity

Overall the manuscript appears to be comprehensive and the data presented valid. However, the behavioral data set should be readdressed and reinterpreted.

R: Please, find below our point-by-point answers.

III. Figure-by-Figure Remarks

Below are remarks of things that should be addressed:

Figure 1 and related: Clinics and virus

Line 104-106: The claim that "female hamsters also exhibited important body weight loss but milder clinical scores compared to males (Fig.1E, F)" should be supported by statistical comparisons (e.g. direct comparisons of clinical scores between sexes).

R: We added statistical information in Figure 1 and in the text (lines 108-113).

Figure legends should show what is presented in the panels in terms of mean/SEM for B and E and the median or mean (?) for C and F. Showing some kind of range would be nice for the score data.

R: The data are expressed as median and interquartile range. We have added this information in all figure legends.

Lines 112-114: The statement regarding weight gain from 10 dpi to 80 dpi, where infected males did not reach the weight of the control group, requires appropriate statistical support.

R: We added statistical information in Figure 1 and in the text (lines 108-113).

Lines 116-120: The authors note a higher LW/BW ratio in some groups, suggestive of inflammation, edema, and congestion. It should be made clear that this was only a trend for the Wuhan group as shown in the figure.

R: We completed the LW/BW ratio graph with data from 4 dpi and moved it to Figure Sup. 1. We also rewrote the related sentence (lines 135-137).

Figure 2 and related: Histology

The authors show and describe myeloid and astrocytic changes in different brain regions. Astrocytic changes observed at 80 dpi (and 4 dpi) are highly interesting as there are as yet no data sets showing such late time points (Lines 172-177). I would like to see some kind of quantification to warrant that this is a true effect and if perhaps there is a sex effect as well. The acute Iba1 effect has already been described in the literature and is fine this way describing it.

R: We quantified Iba1 and GFAP according to dpi, sex and brain region. To facilitate reading, we kept GFAP data in Figure 2 and moved Iba-1 data in Supplementary Fig 5.

Figure 3-5 and related: RNAseq

This is a highly valuable dataset for learning about and deciphering what happens post SARS-CoV-2 in the hamster brain and the presented changes are highly interesting. The way the RNAseq data is presented is great this way, but the field would benefit from an additional look at the sex differences that are already spread all over the paper, but not at all for this data set. The $n = 4$ per sex should allow at least to a certain degree to see if males and females respond differently to the infection in the acute and long-term phase.

R: We added a plot showing the sex-specific information on the RNAseq data in Figure Sup. 6. The gene list for each comparison is included in the Source data excel sheet.

Figure 6 and related: Behavior

Behavioral Data – Novel Object Recognition (NOR)

The raw NOR data do not seem to fully support the claim of persistent memory impairment in both Wuhan-infected males and females (lines 332-333). Specifically: Supplementary Figure 11 shows that Wuhan-female hamsters have a significantly decreased discrimination index at day 79, while other female groups are similar to mock controls.

In male hamsters, there is only a trend for decreased DI at 79 dpi, though an early effect is noted.

Panels D and G of Figure 6 appear identical between males and females, which raises concerns regarding possible misplacement of panels given that supplemental data suggest sex differences.

R: The reviewer is absolutely right in pointing out that the raw data from Supplementary Figures 10-13 do not perfectly match their modeled counterparts in the main text (Figure 6). In the specific case of NOR, Supplementary Figure 11 (now Sup Fig 12) indeed reveals that female hamsters infected with Wuhan at 79 dpi have consistently significantly lower discrimination index than the Mock animals. The temporal trend seems to appear late (after 31 dpi), whereas in males that difference can be observed at 17 dpi (significant difference $P = 0.0074$) and is preserved, although not reaching statistical significance anymore, across 31 dpi and 79 dpi.

The statistical interpretation would be that there is some degree of interaction between the effect of time post-infection and sex. However, we found that the corresponding mixed-effect models did not perform better than the simple, additive model, in terms of goodness-of-fit. In turn, Figure 6 presents the results from the most parsimonious model where that differential

effect was not retained. We have therefore toned down our interpretation in the main text in lines 367-370. We also added the complete statistical analysis in lines 872-873.

Behavioral Data – Hyponeophagia and Sucrose Splash Test

Hyponeophagia:

In the raw data (Fig Sup 8) we see a clear effect in males, but not in females. While males at 16 dpi show reduced anxiety for Delta (and a trend for such a decrease for Omicron) Wuhan males are not different from mock animals. There is a clear increase at 78 dpi for the Wuhan animals.

Looking at Panels B and E of Figure 6 suggests that females show an anxiety phenotype at 78 dpi which is not seen in the raw data and hence may not be reliable, while the male 78 dpi effect for Wuhan is credible.

R: Regarding the behavioral data (hyponeophagia), we have found upon re-examining the data that three-way interactions fitted the data more accurately and therefore thank the reviewer for thoroughly examining the raw data. As a result, the misleading subtle, yet significant increase in anxiety previously observed in Wuhan-infected female hamsters is not visible anymore (Fig. 6). We also added the complete statistical analysis in lines 872-873.

Sucrose Splash Test:

The text (lines 325-328) should be rephrased for clarity, after mentioning a dichotomy I would expect to read about the differences between males and females? The term “depressive-like behavior” should be maintained rather than implying clinical depression in hamsters.

R: We rewrote this section according to your suggestions (lines 358-362).

IV. Significance

3. Significance

Novelty and Impact:

The persistent presence of viral RNA in the brain up to 80 dpi is a novel observation that contrasts with other studies reporting viral clearance within days or weeks (e.g. Frere et al.). This finding may offer insight into potential mechanisms underlying long-term neurological and behavioral changes.

R: We added detection of viral proteins to strengthen the data about persistent infectious virus (Fig. 1, Supplementary Fig. 3).

The extensive dataset—ranging from molecular to behavioral measures—provides valuable information for the field. However, given that several aspects of the data (e.g., the behavioral readouts) are interpreted with advanced statistical models that do not always align with raw data trends, further validation is needed before drawing broad conclusions.

R: We added the complete statistical analysis in lines 872-873. As we stated in lines 334: we used mixed model regression to account for within-individual correlation arising from repeated measurements.

The behavioral outcomes are interesting and after getting them into a better shape will hopefully reveal an interesting long-term phenotype of “post covid hamsters.”

R: We revised the statistical analysis of the behavioral tests and made it available (lines 872-873).

V. Data and Methodology

Figure Presentation and Statistical Details

Figure Legends:

Clarify the statistical presentation in Figures 1 and 6. For instance, in Figure 1, panels B and E should indicate whether values are represented as mean \pm SEM, and panels C and F should

specify whether median values or means are used, ideally including a range for clinical score data.

R: We completed the legends of Figures 1 and 6 with the requested information.

Cohort Consistency and Controls

It is unclear if PBS/mock controls were tested simultaneously with each strain or pooled from different experimental batches. The manuscript should state whether experiments with Wuhan, Delta, and Omicron/BA.1 were conducted concurrently. If not, the methods (and potentially the supplementary figures) should account for this (e.g., via a 2-way ANOVA testing for both cohort and virus effects).

R: Infected and mock-infected animals were tested simultaneously. We added this info in lines 746-749.

Behavioral Testing Conditions

The methods should specify the lighting scheme (e.g., light intensities in lux in the center vs. border of the NOR arena), the time of day when tests were conducted, and report the velocity/distance moved by hamsters during tests. This information is critical, as hypo- or hyperactivity can influence behavioral outcomes.

R: The behavior tests were performed during the 14-hours of light (lines 749-750). The velocity/distance moved by hamsters was not recorded as these are not readouts of the NOR test. Nevertheless, we have previously described that SARS-CoV-2 infected animals do not present signs of locomotor deficit (de Melo et al., 2021).

de Melo, GD et al. COVID-19-related anosmia is associated with viral persistence and inflammation in human olfactory epithelium and brain infection in hamsters. Science Translational Medicine 13, eabf8396 (2021).

Additional video-based analyses (e.g., time spent in center vs. border zones) could help distinguish between memory impairment and anxiety-related behaviors, especially for the NOR video dataset.

R: We thank the reviewer for this interesting comment, but we did not record these points as they are not readouts of the NOR test. These are readouts related to the Open field test.

Code and Statistical Transparency

Given the use of advanced statistical analyses (e.g., mixed model regression), the authors should provide the analysis code in a supplementary file. Additionally, a supplementary table summarizing effect sizes and/or p values would aid in evaluating the robustness of the findings.

R: The mixed-effect model analyses are available in the Git repository: <https://gitlab.pasteur.fr/hub/19421-long-covid-brainstem>. We added this information in the "data availability" section (lines 872-873).

VI. Analytical Approach

Strength of Statistical Methods:

The RNAseq data presentation is well done, but the analysis would benefit from an exploration of sex-specific responses, especially given the $n = 4$ per sex in both the acute and long-term phases.

R: We added more info on sex- and days post infection-related data in Supplementary Fig. 6.

The advanced statistical methods (mixed model regression) used to account for inter-animal variance in the behavioral tests are appropriate; however, the discrepancies between these modeled data and the raw data (particularly in the NOR test) necessitate a re-examination of the analysis approach to ensure that findings are not overstated.

R: Overall, we have re-examined all regression models and added the code in the Git repository: <https://gitlab.pasteur.fr/hub/19421-long-covid-brainstem>.

VII. Suggested Improvements

Statistical Reporting:

Include direct statistical comparisons between male and female groups where claims are made (e.g., body weight changes, clinical scores).

Clarify and standardize the presentation of data in figures, explicitly stating whether the data are means, medians, ranges, etc.

R: We added statistical comparisons throughout the text and in the graphs. The type of data presentation is stated in the legend of each figure.

Provide a supplementary table of effect sizes/p values for the behavioral analyses.

R: Info added the code in the Git repository: <https://gitlab.pasteur.fr/hub/19421-long-covid-brainstem>. (lines 872-873)

Data Consistency and Validation:

Revisit the NOR data analysis (and all the data to be sure) to resolve the inconsistencies between raw data (including supplementary figures) and the main figures. Double-check Figure 6, Panels D and G to ensure the correct data is presented for each sex.

R: We have re-examined all regression models and added the code in the Git repository: <https://gitlab.pasteur.fr/hub/19421-long-covid-brainstem>. Panels D and G present the correct data. As stated before, Figure 6 presents result from the most parsimonious model where that differential effect was not retained.

Include quantification of astrocytic changes (at 4 dpi and 80 dpi) to confirm the reported effects and assess potential sex differences.

R: We added the quantification of astrocytic changes and show that there is no sex-related difference (Fig. 2).

Additional Experimental Suggestions:

It would strengthen the manuscript to include histological validation (e.g., immunostaining) of the mRNA changes at 80 dpi for markers relevant to neurodegeneration or inflammation (e.g., TH, alpha-Synuclein, dopamine or glutamate receptors, IFN-related proteins).

R: We thank the reviewer for this suggestion, but histopathologic validation was beyond the scope of our study. We decided to corroborate the RNAseq data by quantifying the expression of some genes by RT-qPCR (Supplementary Figures 7-8).

An additional deeper look at the RNAseq data in terms of sex effects/differences at 4 and 80 dpi.

R: We added this information on Supplementary Fig 6.

Velocity or distance moved of the hamsters during the behavioral tests should be analyzed and shown as the activity of the animals largely influences the outcomes of the tests (e.g., performers and non-performers, do animals of certain groups explore less or more due to hypo- or hyperactivity?).

R: We did not record these points as they are not readouts of the behavior tests that we used. Nevertheless, we did not notice locomotion issues in the hamsters during the tests.

Light intensities for the behavioral tests and zones need to be reported (xxx lux in center of the NOR arena, xxx lux in the border zone etc.).

R: Light supply was added to increase lighting in the center of the arena (~800 lux) (lines 759 and 760). Lighting was adjusted for the tests for anxiety (light/dark box test and the novelty-suppressed tests), but not for Novel object recognition test.

Methodological Details:

Specify whether all behavioral tests were performed in the same experimental batch and under identical conditions, e.g., were the different virus groups different batches?

R: Due to logistical reasons (limited number of BSL-3 isolators), it was not possible to run all different virus groups experiments at the same time. Nevertheless, in the behavioral tests, there was always a mock-infected group tested parallelly to an infected group.

Were the PBS controls in the same experimental tests at the same time? Were they pooled in the analyses from different batches?

R: The tests with infected and mock-infected animals were performed simultaneously (lines 745-749).

If the Wuhan, Delta, and Omicron experiments were not performed at the same time, this should be stated in the methods section. The supplementary figures should show this somehow, as the mock controls are currently pooled (e.g., a 2-way ANOVA could be used for testing cohort and SARS-CoV-2 effects if the data allows this in terms of distribution).

R: The tests with infected and mock-infected animals were performed simultaneously (lines 745-749). We also added detailed information in the Source data excel sheet.

Detail the lighting conditions, light scheme, and test timing, as these factors can significantly impact hamster behavior.

Terminology:

Maintain the use of “depressive-like behavior” rather than suggesting clinical depression in hamsters.

R: We changed this term throughout the text.

Clarify the terminology referring to “parkinsonism” versus “Parkinson’s disease.”

R: We opted to use “Parkinson’s disease” throughout the text.

Code Availability:

Provide the code used for statistical analyses of the behavioral data (regression) in a supplementary file to facilitate transparency and reproducibility.

R: The code was added in the Git repository: <https://gitlab.pasteur.fr/hub/19421-long-covid-brainstem> (lines 872-873).

7. Clarity and Context

Manuscript Accessibility

The manuscript is generally accessible but would benefit from additional context in several areas:

Clear differentiation between trends and statistically significant effects.

R: We removed the trends from the text and figures to increase readability and facilitate interpretation.

Detailed figure legends that allow the reader to interpret mean/median values, SEM, ranges, and effect sizes.

R: We added these details throughout the text, figures and figure legends.

A more thorough description of the behavioral testing environment (e.g., lighting, time of day, arena conditions) to help contextualize the findings.

R: The lighting conditions of the animal facility were added in lines 553-554. The information on the time of the day was added in lines 749-750. The characteristics of the arena are already described in the section of each test.

Context within the Literature

Provide a discussion that situates the persistent brain virus findings within the broader literature, including contrasting studies where viral persistence was not observed.

R: We rewrote the paragraph and discussed this point in lines 431-434.

Consider discussing the role of SNCA in SARS-CoV-2 infection and neurodegeneration, either by referencing existing literature or by indicating if the RNAseq data provide insights.

R: We decided to not include the SNCA gene in the discussion because it was not dysregulated in our RNAseq data.

Some of the changes observed in the RNAseq dataset are in line, others contradicting to the 31 dpi dataset of Frere et al. It would be good to see more discussion on this.

R: The dataset of Frere et al. did not contain the same regions neither the same timepoints as our datasets. Even though, we discussed more about this data in lines 439-442 and throughout the discussion section.

X. Conclusion

The study presents valuable and novel findings using a well-established hamster model, particularly regarding long-term viral persistence and sex-dependent differences. However, several aspects of the data presentation and statistical analysis need to be clarified and strengthened before the manuscript can be considered for publication. I recommend that the authors address the points listed above, providing additional statistical details, clarifying methodological conditions, and ensuring consistency between raw and analyzed data.

R: We thank the reviewer for these comments. We sincerely hope that the revised text now addresses all the points raised by the reviewer.

The discrepancies between the raw data and the regression-modeled data raise concerns about the overstatement or validity of some findings (e.g., for the NOR). While the NOR data warrants skepticism and may require alternative presentation, calculation, and double-checking, the behavioral experiments conducted in a BSL3 facility with hamsters represent a significant achievement given the inherent complexities. Moreover, the RNAseq dataset from the 80 dpi timepoint of the brainstem is of high interest to the community.

R: We used mixed model regression to account for within-individual correlation arising from repeated measurements to properly quantify the effect of covariates (sex, time post-infection, SARS-CoV-2 variant). The mixed-effect model analyses are available in the Git repository: <https://gitlab.pasteur.fr/hub/19421-long-covid-brainstem>. We added this information in the "data availability" section (lines 872-873).

The RNA-seq data are available in the Array Express database (<https://www.ebi.ac.uk/biostudies/arrayexpress>): 4dpi: ArrayExpress E-MTAB-14779; 80 dpi: ArrayExpress E-MTAB-14780 (lines 869-871).

The presentation of data without explicit p values, relying on reader estimation through CI overlaps, poses a challenge. A supplementary table with effect sizes or p values could enhance trust in the model.

R: The p values were incorporated into all figures where they were found to be statistically significant, except for Figure 6. The behavioral analysis contains such a high number of p values that it is impossible to construct a comprehensible figure. To address this, the complete results are accessible in the Git repository at <https://gitlab.pasteur.fr/hub/19421-long-covid-brainstem> (lines 872-873).

Point-by-point answers to the Reviewers

Reviewer #1 (Remarks to the Author):

The authors have addressed some of my points, however, I strongly suggest to add a limitations section to the discussion, summarizing the limitations discussed with the reviewers. The lack of behavioral effects may be explained by $n=4$, which is way too low for these kinds of behavioral tests with hamsters, and not by the variant. N for all groups should always be stated in the figure legend.

R.: We added the N for the groups in all figure legends. For the behavioral tests (Figure 6), we included a clear description of the N used in each time point: Behavioral session 1 (mock-infected: $n=20$ males + 18 females; Wuhan: $n=12$ males + 12 females; Delta: $n=4$ males + 4 females; Omicron/BA.1: $n=4$ males + 4 females). Behavioral session 2 (mock-infected: $n=20$ males + 18 females; Wuhan: $n=12$ males + 12 females; Delta: $n=4$ males + 4 females; Omicron/BA.1: $n=4$ males + 4 females). Behavioral session 3 (mock-infected: $n=12$ males + 12 females; Wuhan: $n=8$ males + 8 females; Delta: $n=4$ males + 4 females; Omicron/BA.1: $n=4$ males + 4 females). We also acknowledged the limitation on the number of animals in the discussion (lines 364-366).

Reviewer #2 (Remarks to the Author):

The manuscript has been substantially improved in accordance with this reviewers' suggestions. The major concerns previously raised—particularly those regarding viral detection and isolation from the brain, as well as the quantification of astrocytic (GFAP) and microglial (Iba-1) activation, have been adequately addressed. In addition, the text has been thoroughly revised, resulting in a clearer and more cohesive presentation. From this reviewer's perspective, the manuscript is now suitable for publication.

R.: We thank the reviewer for this positive evaluation.

Reviewer #3 (Remarks to the Author):

Dear Authors,

Thank you for your diligent work on the revised manuscript. Your comprehensive responses to the reviewers' comments have significantly enhanced the manuscript's clarity, robustness, and impact. We particularly commend your efforts in providing new immunofluorescence data, quantifying glial changes, refining terminology, and making statistical analyses publicly available.

R.: We thank the reviewer for this positive review.

There is one minor point regarding Figure 6 (Long-term impact of SARS-CoV-2 infection on neuropsychiatric and cognitive behavior) that we believe could benefit from a bit more clarification for the readership.

- Figure 6 Model Clarification: Panels B, C, D, E, F, and G of Figure 6 display average predicted values derived from mixed-model regressions. To ensure maximum transparency and prevent any potential misinterpretation or questions from readers, it is essential to explicitly state the specific mixed-effect model used for each behavioral test within the Figure 6 legend, or directly on the relevant panels themselves.

- For the Novelty-suppressed feeding test (Panels B and E), the model used was $\text{Day} * \text{Sex} * \text{Infection}$.

- For the Sucrose splash test (Panels C and F), the model used was $\text{Day} * \text{Sex} * \text{Infection}$.

- For the Novel object recognition test (Panels D and G), the model used was $\text{Day} + \text{Sex} + \text{Infection}$.

- Clarifying these precise model specifications will provide readers with a clearer understanding of the data presentation, especially noting how factors like sex and time are incorporated into the predictions. We also recommend explicitly referring to the Git repository (<https://gitlab.pasteur.fr/hub/19421-long-covid-brainstem>) as the source for the complete

statistical analyses and model specifications in the legend. This will ensure full transparency and further underscore the rigor of your statistical approach. Once this small but important clarification for Figure 6 is incorporated, the manuscript is well-prepared for final submission.

R.: We agree with the reviewer and completed the legend of the Figure 6 with the requested information.

Minor detail: p values in the manuscript should be written with a “.”.

R.: We corrected the p values throughout the text.